# Metabolic engineering of doxorubicin biosynthesis through P450-redox partner optimization and structural analysis of DoxA

Arina Koroleva [1,7], Erika Artukka[1,7], Keith Yamada [1,7], Sean A. Newmister [2,7], Ralph J. Harte [2], Hannah Boesger [2], Mikael Londen[1], Jacob N. Sanders [3], Heli Tirkkonen [1], Matti Kannisto[1], Rosan C. M. Kuin [4], Mandy Hulst [5], Rongbin Wang [1], Ester Leskinen[1], Morgane Barillec[1], Jarmo Niemi [1], Gilles P. van Wezel [5], Jacques Neefjes [4], S. Eric Nybo [6], Kendall N. Houk [3], David H. Sherman [2], Robbert Q. Kim [4] & Mikko Metsä-Ketelä [1] ✉

Doxorubicin, a widely used chemotherapy drug, is produced by *Streptomyces peucetius* ATCC27952. The biosynthesis relies on the cytochrome P450 monooxygenase DoxA, which catalyzes three consecutive late-stage oxidation steps. However, conversion from daunorubicin to doxorubicin is inefficient, necessitating semi-synthetic industrial manufacturing. Here, we address key limitations in DoxA catalysis. We identify the natural redox partners ferredoxin Fdx4 and ferredoxin reductase FdR3 by transcriptomic analysis. We discovered the vicinal oxygen chelate family protein DnrV to prevent product inhibition by binding doxorubicin. Structural analysis of DoxA and density functional theory (DFT) calculations reveal that inefficient C14 hydroxylation results from the unfavorable anti-conformation of the methyl ketone side chain of daunorubicin. We harness these advances for rational strain engineering, leading to an 180% increase in doxorubicin yields and an improved production profile. This study provides singular insights into enzymatic constraints in anthracycline biosynthesis and facilitates cost-effective manufacturing to meet the growing global demand for doxorubicin.

Anthracyclines are an important class of microbial natural products that have been a cornerstone in clinical anticancer chemotherapy for several decades[1]. These molecules consist of a linear tetracyclic 7,8,9,10-tetrahydro-5,12-naphthacenoquinone scaffold, which is decorated with one or more carbohydrate moieties[2]. The best-known anthracyclines are daunorubicin (DNR) and doxorubicin (DXR), which are first-choice chemotherapeutic agents administered to more than one million cancer patients each year in the treatment of lymphomas, carcinomas, and sarcomas[3]. The biological activity of anthracyclines is mediated primarily via interactions with DNA, where the polyphenolic tetracyclic aglycone intercalates between base pairs and the carbohydrate unit at C7 is positioned in the minor groove of DNA[1]. Intercalation to DNA promotes poisoning of topoisomerase II, leading to DNA double strand brakes and histone eviction, both of which lead to cell death[4,5].

[1]Department of Life Technologies, University of Turku, Turku, Finland. [2]Life Sciences Institute, University of Michigan, Ann Arbor, MI, USA. [3]Department of Chemistry, University of California, Berkeley, CA, USA. [4]Department of Cell and Chemical Biology, Leiden University Medical Centre, Leiden, The Netherlands. [5]Institute of Biology, Leiden University, Sylviusweg, Leiden, The Netherlands. [6]Department of Pharmaceutical Sciences, College of Pharmacy, Ferris State University, Big Rapids, MI, USA. [7]These authors contributed equally: Arina Koroleva, Erika Artukka, Keith Yamada, Sean A. Newmister. ✉e-mail: mianme@utu.fi

DNR was originally discovered in *Streptomyces peucetius* ATCC 27952 cultures and later clinically introduced for treating acute leukemia and lymphoma[3]. DXR was identified serendipitously during strain development of *S. peucetius var. caesius*, when a C14 hydroxy derivative of DNR accumulated in the culture medium following random mutagenesis of the parental strain[6]. DXR exhibited enhanced cytotoxic activity and a higher therapeutic index than DNR. However, DXR-producing strains typically co-produce substantial amounts of DNR, complicating industrial-scale biosynthetic production. The structural similarity of DXR and DNR poses challenges for chromatographic separation and downstream processing. Consequently, DXR and related anthracyclines in clinical use—such as epirubicin, pirarubicin, and idarubicin—are manufactured semi-synthetically from DNR[1].

The biosynthesis of DNR and DXR has been intensively investigated due to their clinical relevance[2]. Sequencing of the biosynthetic gene cluster (Fig. 1A) clarified that a canonical type II polyketide synthase system generates the aglycone from one propionyl-CoA and nine malonyl-CoA molecules via iterative Claisen condensations (Fig. 1B)[7]. In addition, the biosynthesis of the TDP-daunosamine carbohydrate and attachment at C7 has been fully elucidated[8]. After glycosylation, the tetracyclic core is modified in extensive tailoring reactions, which include C15 demethylation, C10 decarboxylation, C4 *O*-methylation, and C11 hydroxylation[9]. In the context of DXR biosynthesis, the most intriguing steps are the three C-H oxidation reactions catalyzed by a single heme-containing cytochrome P450 enzyme DoxA (Fig. 1B)[8]. The enzyme performs sequential conversion of 13-deoxydaunorubicin (DOD) to 13-dihydrodaunorubicin (DHD), further C13 oxidation to DNR, and finally catalyzes C14 hydroxylation to generate DXR[10]. Enzyme kinetic studies of DoxA revealed that the $V_{max}$ values varied significantly for the three substrates and were 520-fold slower for DNR than for DHD[10]. In addition, early in vivo experiments identified that

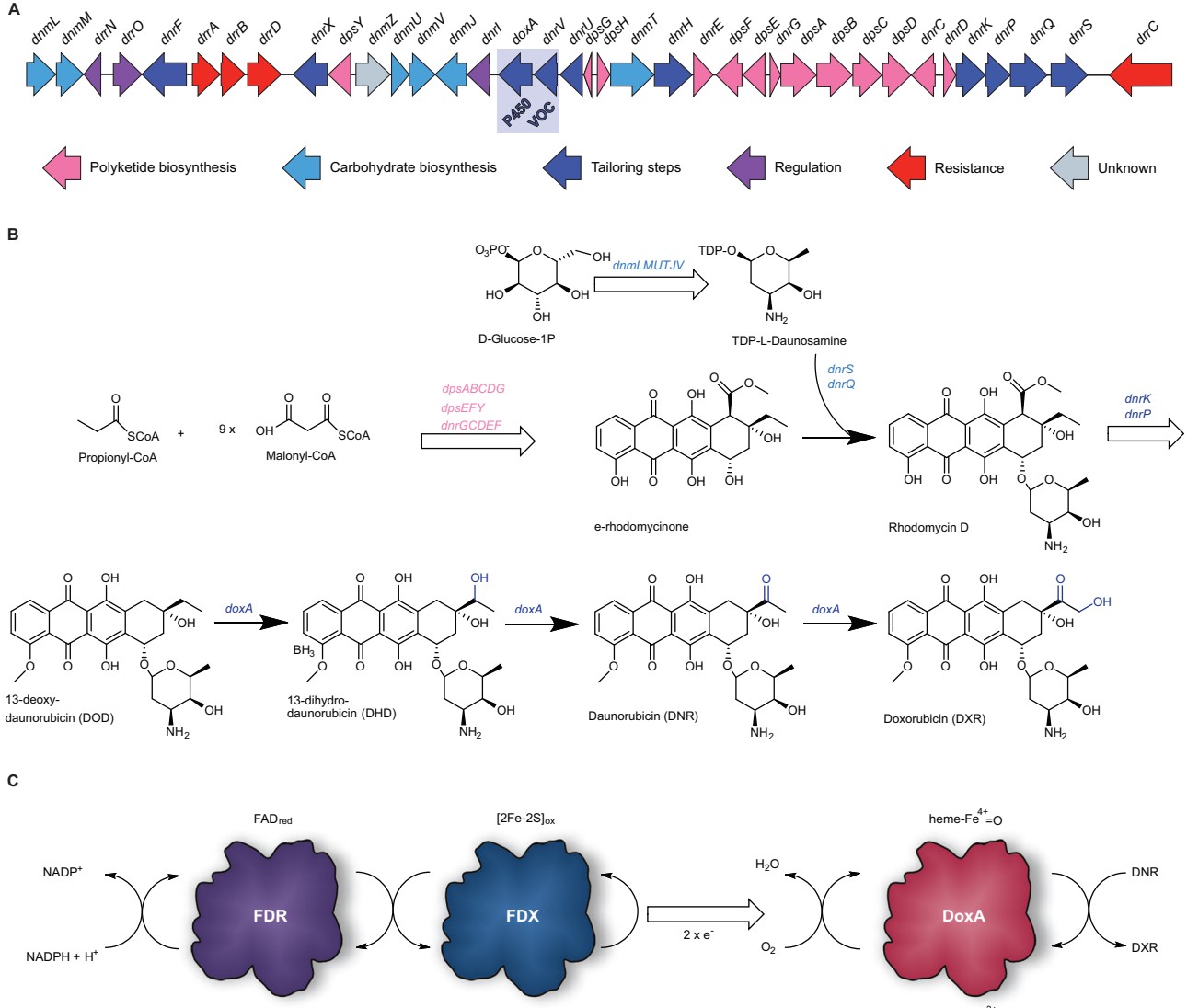

**Fig. 1 | The biosynthesis of daunorubicin (DNR) and doxorubicin (DXR).**
**A** Organization of the daunorubicin biosynthetic gene cluster. The key genes *doxA* and *dnrV* are highlighted in blue and annotated to encode cytochrome P450 and vicinal oxygen chelate (VOC) enzymes, respectively. **B** Abbreviated biosynthetic scheme for formation of DXR. Polyketide biosynthesis results in the formation of ε-rhodomycinone, which is glycosylated by attachment of L-daunosamine, leading to the formation of rhodomycin D. Polyketide tailoring results in the formation of the intermediate 13-deoxydaunorubicin (DOD) important for this study. The P450 enzyme DoxA catalyzes three consecutive hydroxylation reactions that convert DOD to 13-dihydrodaunorubicin (DHD), DNR, and finally to DXR. **C** The redox cofactors of P450 enzymes consist of ferredoxin reductases (FDR) that utilize NADPH to convert oxidized flavin ($FAD_{ox}$) to reduced flavin ($FAD_{red}$). In turn, $FAD_{red}$ is reduced by the [2Fe-2S] centre of ferredoxins (FDX), which enables donation of two single electrons to the heme of DoxA. Reduction of heme-$Fe^{3+}$ allows a reaction with molecular oxygen $O_2$ and formation of a high valent heme-$Fe^{4+}$=O iron oxo species, which is able to oxidize carbon atoms by C-H activation.

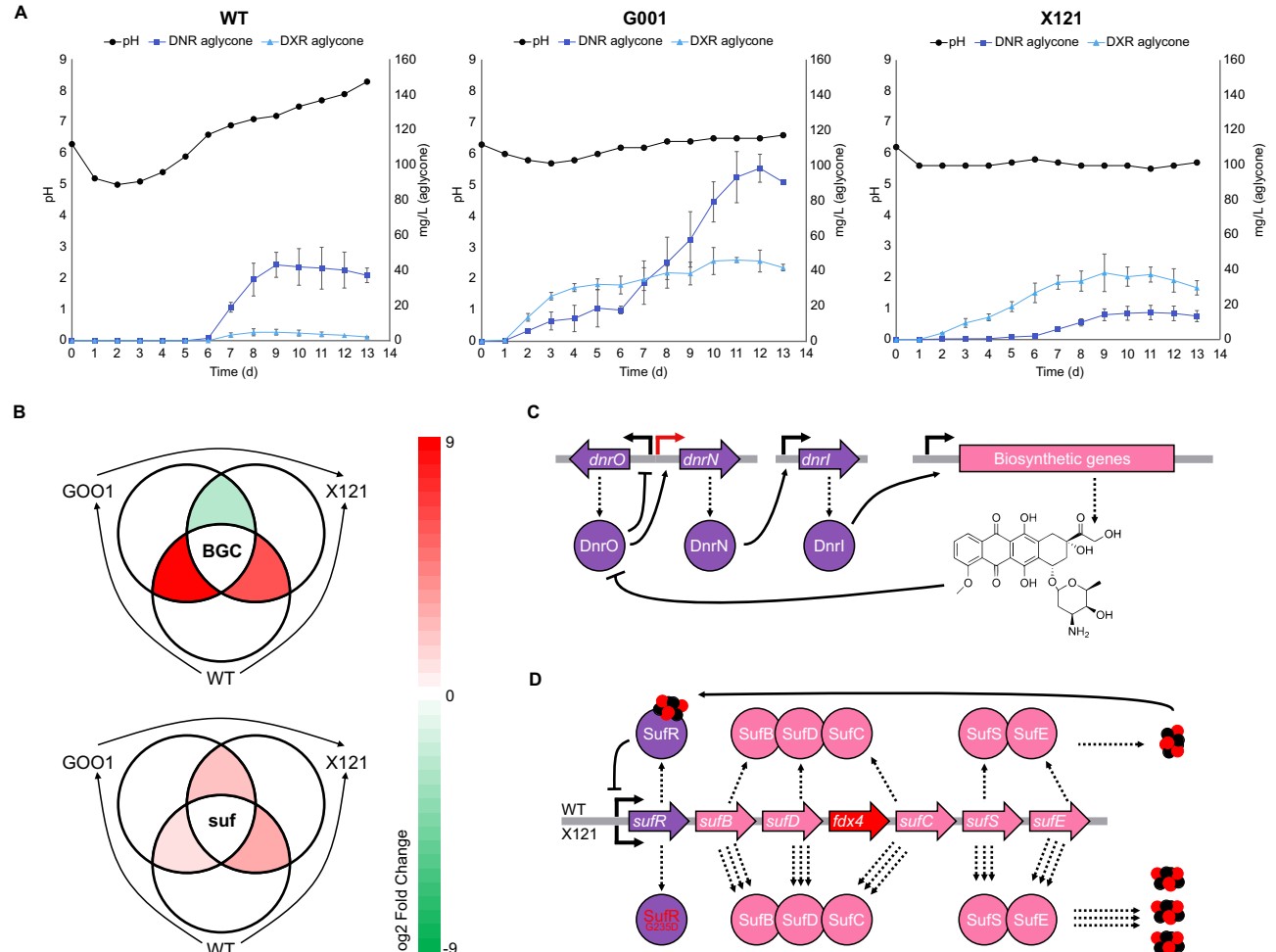

**Fig. 2 | Comparative metabolome, genome, and transcriptome analysis.**
**A** Metabolite production profiles of *S. peucetius* ATCC 27952 wild-type (WT), and mutant strains G001 and X121. The experiments were conducted in triplicate (n = 3 biological replicates). Data are presented as mean values ± SD. **B** Average day 2 differential expression of genes in the *dnr* BGC (top) and *suf* cluster (bottom). **C** Cluster situated regulation of the daunorubicin and doxorubicin BGC. The red

arrow shows the mutation in the bidirectional promoter of *dnrO* and *dnrN*, resulting in the overexpression of *dnrN* and subsequently the overexpression of the BGC, in G001 and X121. **D** Biosynthetic scheme for the formation of Fe-S clusters by the *suf* pathway and upregulation of the ferredoxin *fdx4*. The strain X121 harbored a mutation in the negative regulatory gene *sufR* that leads to activation of the cluster. Source data are provided as a Source Data file.

the gene *dnrV*, encoded directly upstream of *doxA*, positively influences doxorubicin production[8]. DnrV is annotated as a member of the vicinal oxygen chelate (VOC) enzyme family, which is typically not associated with P450 enzymes, and the precise role of this protein in doxorubicin biosynthesis remains unknown.

P450 enzymes have been extensively investigated for their important role in various biocatalytic processes[11]. One of the challenges in engineering P450 systems is the requirement for a complex co-factor-mediated electron transport chain to promote activation of the heme co-factor (Fig. 1C). Classical microbial P450 redox systems are composed of a flavin-dependent ferredoxin reductase (FdR) that utilizes NAD(P)H for the reduction of ferredoxin (Fdx). In turn, Fdx catalyzes the single electron reduction of the heme $Fe^{3+}$ iron to $Fe^{2+}$, which enables a reaction with molecular oxygen and formation of the catalytic high-valent iron-oxo-heme species that is capable of oxidizing unactivated C-H bonds of substrates[12]. The correct pairing of the FdR-Fdx-P450 components is vital for the performance of P450 biocatalysts[13], but identification of the natural redox partners of DoxA has been challenging[14,15]. The genome harbors six *fdx* and seven *fdr* genes, which are not co-localized within the daunorubicin biosynthetic genes. In addition, even though attempts have been made to identify

the redox partners of DoxA by predictive molecular modeling, the improvements in catalysis have been modest[14].

Here, we identify three factors that limit DXR biosynthesis in *S. peucetius* ATCC 27952 and overcome these bottlenecks in strain engineering. The work includes (i) identification of the natural redox partners of DoxA from transcriptomics data, (ii) elucidation of the role of DnrV as a drug-binding protein that improved catalysis by preventing product inhibition of DoxA, and (iii) determination of the ternary crystal structures of DoxA in complex with ligands, which reveal the molecular basis for poor catalytic turnover of DNR. These advances guide our rational metabolic engineering efforts in the creation of a strain with a significantly improved DXR production profile and yield.

## Results and Discussion

### Identification of DoxA redox partners by comparative metabolome, genome and transcriptome analyses

We performed comparative time-course analysis of metabolite production by wild-type *S. peucetius* ATCC27952 and two industrial strains generated by random mutagenesis. The industrial strains included *S. peucetius* G001 and X121 (Fig. 2A), which produce high yields of DNR

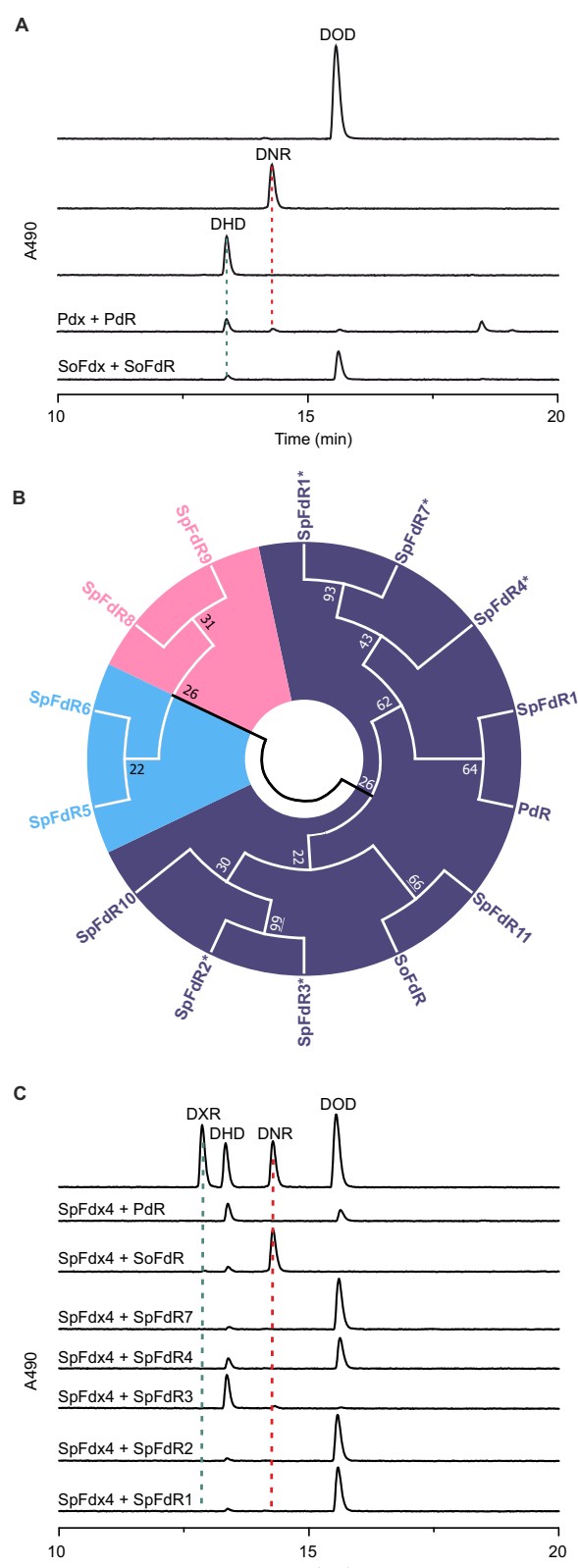

**Fig. 3 | Identification of the natural redox partners of DoxA. A** DoxA reactions with non-native redox partners PDX + PDR from *P. putida* and SFX + SFR from spinach using DOD as a substrate, **B** Phylogenetic tree of putative FDR enzymes, including putidaredoxin-type (purple), adrenodoxin-type (blue), and plant-type ferredoxin reductases (pink) identified from the genome of *S. peucetius* ATCC27952. Reference sequences for putidaredoxin reductase-type (BphA4, *Pseudomonas* sp.; PdR, *Pseudomonas putida*; SmFdR, *Stenotrophomonas maltophilia*), adrenoredoxin reductase-type (AtFdR, *Arabidopsis thaliana*; MtFdR, *Mycobacterium tuberculosis*), and plant ferredoxin reductase-type (ZmFdR, *Zea mays*; SoFdR, *Spinacia oleracea*; PsFdR, *Pisum sativum*) were used to guide the tree. Proteins marked with an asterisk (*) could be produced in soluble form and were used in enzyme activity assays, **C** DoxA reactions with FDX4 and various ferredoxin reductase partners using DOD as a substrate.

contrast, DNR was primarily produced in the late stationary growth phase, particularly in G001 (Fig. 2A).

Genome and transcriptome sequencing data clarified several parameters related to the early onset of anthracycline production in G001 and X121 (Fig. 2B, Supplementary Table 1). The regulation of the DNR biosynthetic gene cluster is under a three-tier transcriptional activator system[1]. The TetR-family transcriptional regulator *dnrO* binds to all glycosylated products of the pathway and controls the activation of *dnrN*. In turn, DnrN is a pseudo-response regulatory protein that governs the expression of *dnrI*, which directly controls the transcription of the biosynthetic and resistance genes by binding to the promoter regions within the *dnr* cluster[17]. Both G001 and X121 harbored a deletion mutation in the bidirectional promoter region of *dnrO* and *dnrN* (Fig. 2C), which may have contributed to early anthracycline biosynthesis. Consistently, transcriptomics data revealed 3.4- and 2.6-fold overexpression of *dnrN* in G001 and X121, respectively, compared to the wild-type on day 1 (Fig. 2C, Supplementary Table 1). Moreover, we identified a mutation in the *drrD/dauW* gene, which has been suggested to negatively regulate DNR production[18]. Finally, ATP-glucokinase[19] was downregulated in the mutants G001 and X121 on day 1, thereby decreasing carbon catabolite repression and promoting early onset of DNR and DXR production.

Analysis of the biosynthetic gene cluster did not provide insight into the production ratio of DNR/DXR. Neither G001 nor X121 had accumulated any mutations in the *doxA* gene, and the upregulation of *doxA* was consistent with the overall overexpression of the *dnr* biosynthetic genes (Fig. 2B, Supplementary Table 1). Therefore, we surmised that the higher DXR yields DXR in G001 and X121 could be indirectly influenced by the earlier onset of production due to differential expression of endogenous *fdx/fdr* partners during the *Streptomyces* growth cycle. Comparative transcriptomics revealed significant upregulation at two genetic loci in X121 compared to G001 and the wild-type (Supplementary Table 1). We observed overexpression of the *suf* pathway[20], which included 3.4-fold upregulation of the ferredoxin *fdx4* and six genes related to iron-sulfur cluster assembly in X121 (Fig. 2B, D, Supplementary Table 1). The regulatory gene *sufR* contained a point mutation leading to a Gly235Asp substitution in X121, possibly resulting in overexpression of the *suf* operon[21]. We inactivated *sufR* in *S. peucetius* ATCC27952Δ*sufR*, which resulted in activation of the *suf* pathway and 4.4-fold overexpression of *fdx4* (Supplementary Fig. 1). The second operon upregulated in X121 encoded the ferredoxin *fdx5* and the *tau* pathway[22] (Supplementary Table 1) responsible for the acquisition of sulfur from taurine. However, we did not consider *fdx5* as the correct redox partner, since transcription of the *tau* pathway was exceedingly low in G001 during high-yield production of DXR in days 2-4. The *fdx/fdr* pairs previously studied in conjunction with DoxA[14] were transcribed at a basal level and did not show differential expression between any strain combinations (Supplementary Table 1).

and DXR, respectively. Anthracycline production is under carbon catabolite repression in the wild-type strain[16] and, in agreement, we observed the initiation of DNR production on day 6. In contrast, anthracycline production could already be observed during the exponential growth phase on day 2 in G001 and X121. Both G001 and X121 produced DXR mainly in the exponential and early stationary growth phases, reaching DXR yields of approximately 40 mg/L. In

## The ferredoxin SpFdx4 and ferredoxin reductase SpFdR3 are the natural redox partners of DoxA

To evaluate the various SpFdx and SpFdR redox partners from *S. peucetius* ATCC27952, we established an in vitro enzymatic activity assay. DoxA was expressed and purified from *E. coli* (Supplementary Fig. 2), and an active heme prosthetic group was confirmed spectrophotometrically (Supplementary Fig. 3). Under optimized conditions, spinach *Spinacia oleracea* ferredoxin reductase (SoFdR) and spinach ferredoxin (SoFdx) converted a small amount of DOD to DHD (11%) based on HPLC analysis (Fig. 3A), while putidaredoxin reductase PdR and putidaredoxin Pdx from *Pseudomonas putida* performed better with the production of both DHD (69%) and DNR (18%) (Fig. 3A). All activity assays required rigorous optimization to determine the correct ratio of the P450-Fdx-FdR components, since an excess of ferredoxin reductases e.g. PdR led to reductive deglycosylation (Supplementary Fig. 4A). As noted previously[23], the deglycosylation reaction is dependent on direct reduction of anthracyclines by ferredoxin reductases, which we observed with DHD and PdR even in the absence of Pdx and DoxA (Supplementary Fig. 4B). Preparative scale reactions with DNR and SoFdR verified the reaction product as 7-deoxydaunorubicinone (Supplementary Table 2, Supplementary Figs. 5-10), which is likely formed from DNR under anoxic conditions via a quinone methide intermediate (Supplementary Fig. 4C)[23].

To study native redox partners from *S. peucetius* ATCC27952, recombinant SpFdx4 from the *suf* cluster was first produced and purified from *E. coli*. Based on sequence analysis (Supplementary Fig. 11), SpFdx4 is a Rieske-type $Cys_2His_2Fe_2S_2$ ferredoxin containing two histidine and two cysteine residues for coordination of the [2Fe−2S] cluster, similarly to BphA3[24] and TDO-F[25] ferredoxins affiliated with biphenyl and toluene dioxygenase systems, respectively. Therefore, SpFdx4 is not evolutionarily related to the $Cys_4Fe_2S_2$ plant-type SoFdx[26] or adrenodoxin-type Pdx[27]. Spectrophotometric analysis revealed canonical UV/Vis maxima for SpFdx4 and Pdx (Supplementary Fig. 3B), including expected differences reported for the two ferredoxin subfamilies[28,29].

Our comparative transcriptomics data did not reveal obvious FdR candidate proteins, presumably due to sufficient FdR activity at basal expression levels in the industrial strains. Genome mining and phylogenetic analysis revealed that *S. peucetius* ATCC27952 harbored five putidaredoxin reductase-type and two adrenodoxin reductase-type enzymes, but no plant-type ferredoxin reductases were detected (Fig. 3B)[30]. Five of the seven SpFdR proteins could be produced in soluble form in *E. coli* (Supplementary Fig. 2), which allowed the use of SpFdR1, SpFdR2, SpFdR3, SpFdR4, and SpFdR7 in comparative activity assays with SpFdx4, DoxA, and DOD as a substrate. One of the proteins, SpFdR3, mediated efficient conversion of the DOD substrate into DHD (88%) and DNR (8%) (Fig. 3C). By contrast, low amounts of DHD formation (4–22%) was detected with the other four candidate reductase proteins SpFdR1, SpFdR2, SpFdR4 and SpFdR7 (Fig. 3C). Interestingly, the non-native ferredoxin reductase SoFdR converted nearly all of the substrate DOD to DNR when paired with SpFdx4 and DoxA (Fig. 3C).

## DnrV is an anthracycline resistance determinant that aids DoxA catalysis by preventing product inhibition

None of the reactions resulted in the formation of DXR, which is in line with previous reports on the in vitro activity of DoxA, where only trace quantities of DXR were observed[14,15]. Hence, we turned our attention to *dnrV*, which is located in the same operon as *doxA* and encodes a protein belonging to the vicinal oxygen chelate (VOC) enzyme family[31]. Addition of purified DnrV to reactions with SpFdx4, SoFdR, and DoxA with any of the substrates DOD, DHD or DNR led to concentration-dependent improvement in catalysis (Fig. 4A). We achieved full conversion to DXR without formation of deglycosylated shunt products through use of stoichiometric DNR: DnrV concentrations up to 100 μM DNR substrate (Fig. 4B).

The improved activity assay and the discovery of the beneficial effect of DnrV led us to revisit previous observations that DoxA catalysis is limited by product inhibition[10]. We investigated the influence of DXR in catalysis by titration of the DoxA reaction with DXR (0 μM–10 μM), which revealed concentration-dependent reduced activity (Fig. 4C). In the absence of DXR, a control reaction using DOD as a substrate led to the formation of 59% DXR and 41% DNR, but addition of 10 μM DXR resulted in decreased activity, and the main products were 49% DHD and 30% DNR.

Proteins from the VOC enzyme family have not previously been associated with P450 enzymes, and we hypothesized that DnrV may have an indirect role in enhancing DoxA activity. We observed that DnrV shares sequence similarity with mitomycin-resistant determinant (MDR)[31,32], which is a non-metalloprotein member of the VOC superfamily. This indicated that DnrV may be an intracellular anthracycline drug-binding protein. In agreement, we performed binding studies exploiting the fluorescent properties of anthracyclines to demonstrate that DnrV binds DOD, DNR and DXR at low micromolar $K_D$ values (Fig. 4D). As a positive control, we measured binding of anthracyclines to DoxA, which also occurred at a low micromolar range for the substrates DOD and DNR, but at lower affinity to the product DXR (Fig. 4E). In a negative control reaction, we demonstrated that titration of SpFdx4 with DXR did not alter the fluorescent properties of the anthracyclines (Fig. 4F). The activity appeared to be specific for DnrV, since the related mitomycin MDR did not bind anthracyclines (Supplementary Fig. 12A) or enhance the catalytic rate of DoxA (Supplementary Fig. 12B). These experiments suggest that DnrV indirectly improves DoxA activity by sequestering DXR, which prevents product inhibition from reducing catalysis by DoxA.

## Structural basis for DoxA catalysis

To investigate the molecular basis for C13 and C14 hydroxylation, we solved the crystal structures of DoxA in complex with DOD, DHD, and DNR (Supplementary Table 3). The structure models reveal a crystallographic dimer (Fig. 5A) of typical class I P450 enzymes[33], and monomers of all three structures align well with rmsd values < 1 Å. In the asymmetric unit, two monomers (Fig. 5A) interact on a site close to the heme group. Although the buried site (approximately 750 Å2) does not indicate a biologically relevant interaction, the region overlaps significantly with the likely proximal site identified for electron delivery to the heme from Fdx (Supplementary Fig. 13A)[34–36].

For each ternary complex, electron density for both the heme and the substrates is well-defined, unequivocally placing these molecules in the structures (Fig. 5B). The substrates bind along the central I-helix (amino acids 249–279, Supplementary Fig. 13B). The active site is sealed off in the substrate-bound state, but the high B-factors in the B-C loop (amino acids 69–83) and the C-terminal loop (amino acids 407–410) indicate flexibility (Supplementary Fig. 13C). This flexibility is most prominent in the F and G helices (amino acids 190–207), where the F helix extends further over the bound substrate contributing to hydrophobic packing.

In all substrate-bound structures, the C13 atom preferred for hydroxylation is situated within 4.4 Å of the heme center. The anthracycline aglycone extends towards the N-terminus of the central I-helix and is surrounded by mainly hydrophobic sidechains (Ala261, Phe260, Gly257), with Tyr253 'capping' off the binding site near the C4 methoxy group (Fig. 5B). Other residues anchoring the aromatic part of the substrate are in the $3_{10}$-helix in the B-C loop region (amino acids 94-96) and Leu189 from the F-helix, while Val305 and Leu309 create a hydrophobic cavity for C14 near the active site (Fig. 5C). The substrate binding mode was investigated through mutation of Val305 to Leu or Ile, which led to loss-of-activity (Supplementary Fig. 13D). The L-daunosamine sugar moiety is coordinated through hydrogen bonds with water molecules and polar residues Lys185 and Thr411, as well as the main chain of Gly82 and a

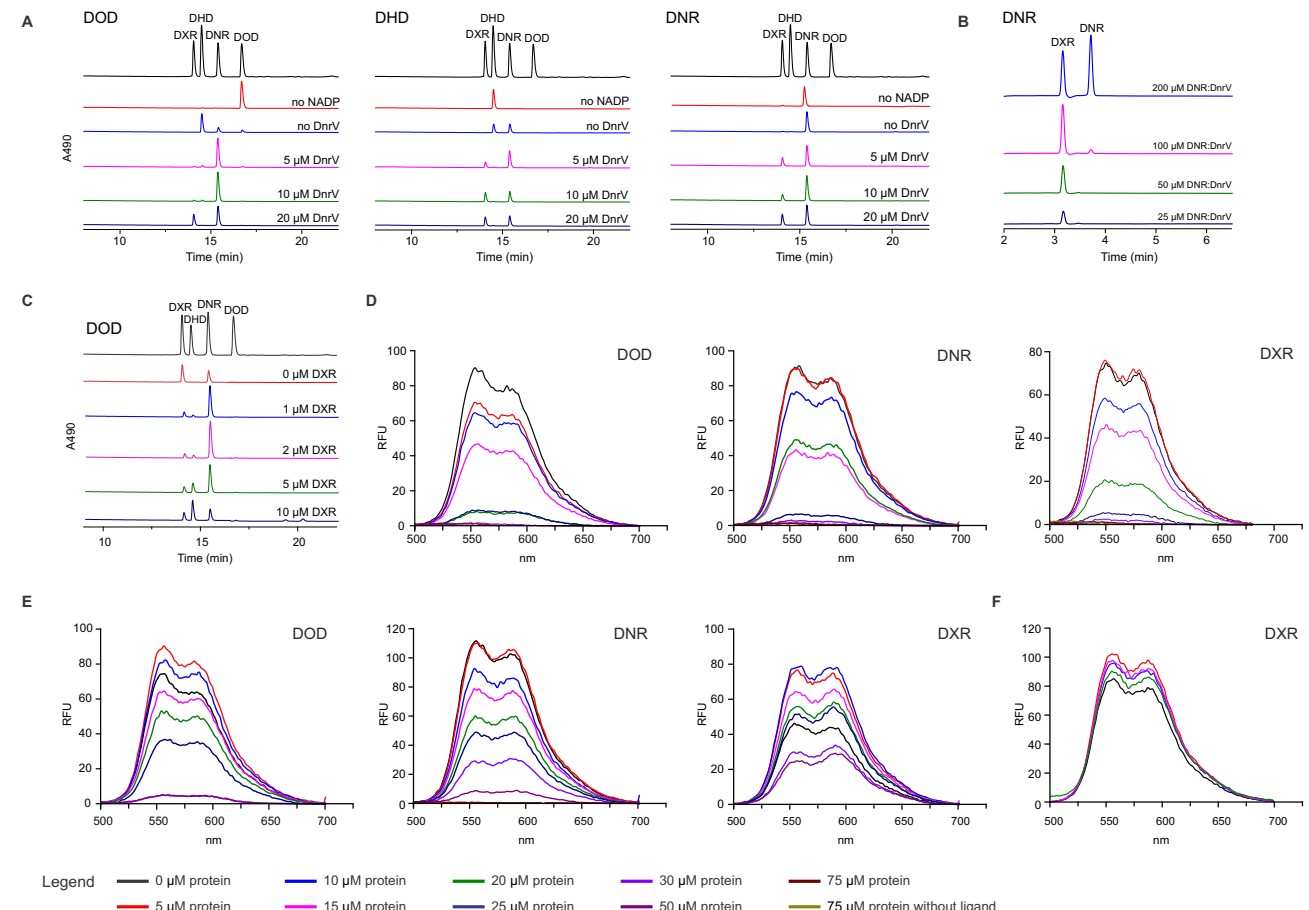

**Fig. 4 | DnrV improves catalysis of DoxA by binding to anthracyclines.**
**A** Increasing the concentration of DnrV improves DXR formation with all three substrates, DOD, DHD, and DNR. Reactions included 10 μM DoxA, 5 μM FDX4, 0.1 μM SFR, 0-20 μM DnrV, 4 mM Glucose, 1U GDH and 2.5 U catalase, 0.5 mM NADP +. No NADP+ was added to the control reactions. **B** Stoichiometric DNR: DnrV ratios enabled full conversion to DXR at 100 μM substrate concentration. This optimized reaction used 20 μM FDX4 instead of 5 μM. **C** Increasing the presence of DXR prevents catalysis by DoxA. Reactions included 20 μM DOD, 7 μM DoxA, 10 μM DnrV, 5 μM FDX4, 0.1 μM SFR, 4 mM glucose, 1 U Glucose dehydrogenase, 0.5 mM

NADP+ and 2 U catalase. **D** Anthracycline binding assay based on quenching of the anthracycline fluorophore upon binding to proteins due to changes in the chemical environment. DnrV binds to the anthracyclines DOD, DNR, and DXR at low micromolar concentrations. **E** DoxA binds to the anthracyclines DOD and DNR at low micromolar concentrations and DXR at higher micromolar concentrations. **F** The ferredoxin FDX4 does not interact with anthracyclines such as DXR. The anthracycline fluorophores were excited at 470 nm, and emission was recorded at 500-700 nm. Source data are provided as a Source Data file.

salt bridge with Asp84 (Fig. 5B). The axial water ligand is well defined in the electron density among the different substrate complexes. This axial water is also near the conserved Thr265 on the I-helix (Fig. 5C), and implicated in proton delivery via the acid-alcohol pair with Asp264 and bridging water molecules that are resolved in the complex structures[33]. Mutation of Thr265 to Ser was highly detrimental to the enzymatic activity in comparison to the wild-type (Supplementary Fig. 13D).

The ternary structures shed light on the relatively high conversion of DOD and DHD to their respective products compared to the low-level catalysis of DNR to DXR. In the structure models, C13 lies in a favorable position with respect to the heme center, while the C14 methyl is oriented away from the heme iron (Fig. 5C). Because the electron density at this resolution does not permit differentiation of the O13 ketone from the C14 methyl of DNR, we performed DFT calculations for both *syn*- and *anti*-conformers of C13, as this rotational barrier could, in part, inform the low rates of C14 C-H hydroxylation by DoxA (Supplementary Table 4). The calculations revealed that the *anti*-conformation between O9−O13 is favored by 2.3 kcal/mol over the corresponding *syn* conformer (Fig. 5D). This computational result is supported by the high-resolution crystal structures of DNR[37] and carminomycin[38,39], where the same *anti*-conformation was observed.

Other factors may contribute to attenuated C14 hydroxylation, namely the energetic barriers for the respective radical intermediates at C13 versus C14, but these do not obviate the requirement for proper substrate orientation in catalysis. Therefore, we performed 1000 ns molecular dynamics simulations with the DOD and DNR complex structures and the iron (IV)-oxo species (Compound I)[33] modeled in place of the heme iron (Supplementary Fig. 14) in triplicate (Supplementary Table 5). These simulations revealed stable substrate binding throughout the simulation, where poses matching the crystallographic orientation were predominant. In the DOD complex, C13 was consistently oriented toward the iron (IV)-oxo (average 43% of frames, Fig. 5E). For DNR, C14 occasionally sampled productive binding poses (average 9% frames) either by rotation about the C13−C14 bond or by repositioning the anthracycline core in the active site (Fig. 5F). Further structural analysis of the MD simulations showed that DNR C13 carbonyl consistently forms a hydrogen bond with the Thr265 sidechain (average 28% of frames, Supplementary Fig. 15), further stabilizing DNR in an unreactive pose for C14 oxidation. Together, these data support the observation that DoxA is highly predisposed for C13 oxidation, as opposed to C14 where low levels of C-H hydroxylation are explained by the structural and computational studies.

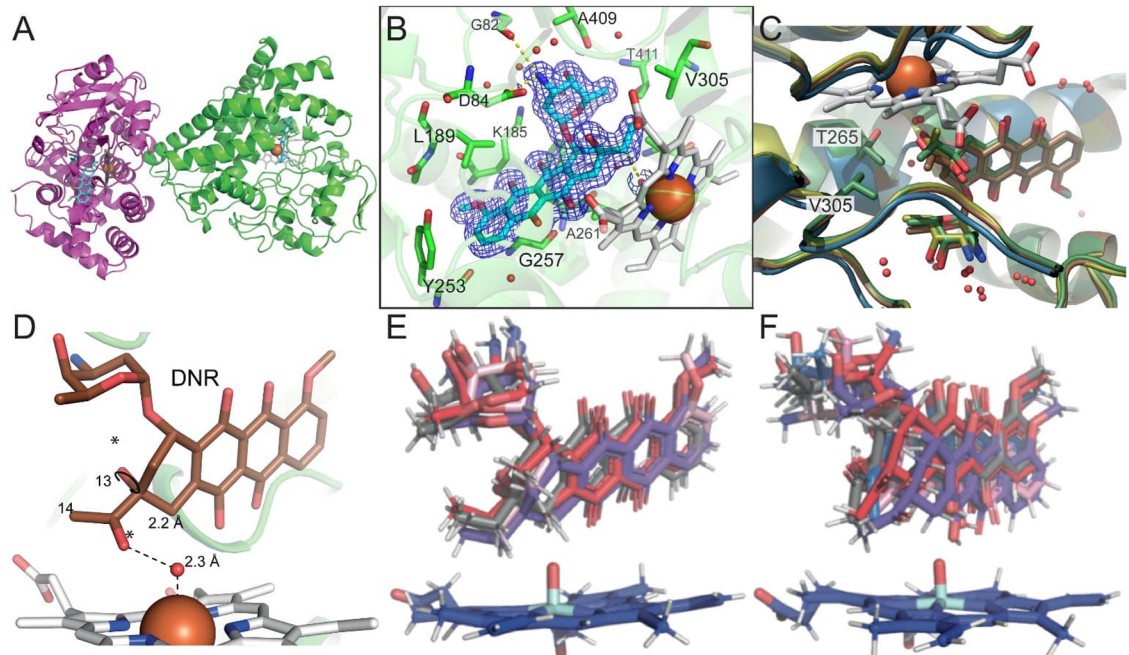

**Fig. 5 | Structural basis for DoxA catalysis. A** The overall structure of DoxA complexed with DOD demonstrates a crystallographic dimer (green, purple) with the heme (grey) and substrate (cyan) stacking perpendicular to each other. **B** The active site of DoxA showing amino acid residues in the vicinity of the substrate DOD, which is contoured with the electron density at 1 σ, highlights the proximity of the substrate to the heme group. **C** Residues Val305 and Leu309 create a hydrophobic cavity for C14, which positions the adjacent C13 carbon close to the

axial water molecule coordinated to the heme. **D** DFT calculations demonstrated an energetic preference for *anti-* conformation. **E** Superposition of the top five most occupied DOD clusters in 1000 ns simulation with DoxA. DOD is stably oriented for C13 oxidation by compound I. **F** Superposition of the top five most occupied DNR clusters in 1000 ns simulation with DoxA. DNR infrequently samples conformation required for C14 oxidation.

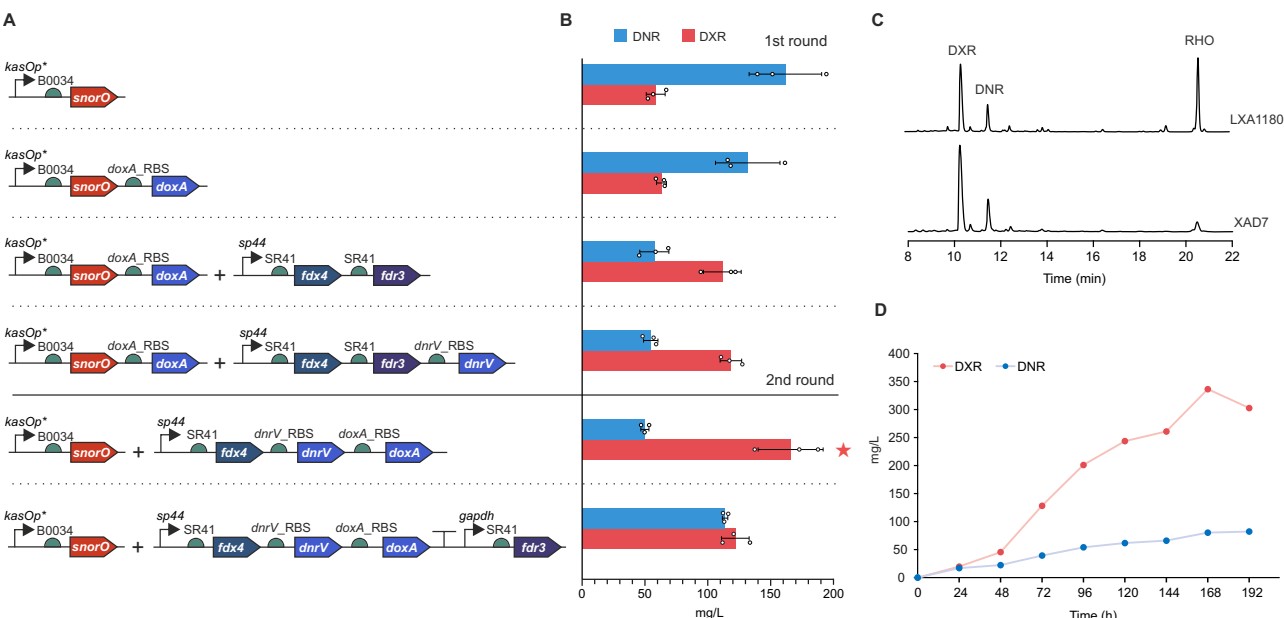

**Fig. 6 | Metabolic engineering of *S. peucetius* G001 for improved production of DXR. A** Schematic presentation of constructs for overexpression of *snorO, doxA, fdx4, fdr3* and *dnrV*. A two-plasmid system was utilized in the expression studies. **B** Production of DNR and DXR by the engineered strains. The experiments were conducted in triplicate (*n* = 3 biological replicates). Data are presented as mean

values ± SD. **C** HPLC production profile of G001/pSnorO+pFdx4_DnrV_DoxA with LXA1180 and XAD7 resins. **D** Time-course analysis for production of DXR and DNR by G001/pSnorO+pFdx4_DnrV_DoxA in a bioreactor. Source data are provided as a Source Data file.

## Metabolic engineering for improved production of DXR

To maximize production of DXR, we introduced all relevant genes into G001 stepwise using two vectors pSET152[40] and pEN-SV1[41] integrating to the *attB* sites of phages ΦC31 and SV1, respectively, under the

control of constitutive promoters *kasOp*[*42], *sp44* and the *gapdh*[43] (Fig. 6A). As the first step, the DNA repair gene *snorO* from the noga-lamycin pathway[44] was introduced under the control of *kasOp** in pEN-SV1 to improve the stability of the strain with initial production of

61 mg/L DXR and 154 mg/L DNR. Notably, the addition of *doxA* alone as a second gene in the same operon did not influence the production profile, indicating that the P450 enzyme was not rate-limiting. The subsequent introduction of the *fdx4-fdr3* redox system as a single operon under the control of *sp44* using pSET152 shifted the metabolic profile towards DXR with a 1.8-fold increase in production of DXR to 112 mg/L. Surprisingly, overexpression of the *dnrV* gene as a third gene in the operon did not significantly improve DXR yields in the first development cycle.

In a second development round, we considered the importance of the correct stoichiometry of the Fdx-FdR-P450 partners for efficient catalysis in the in vitro reactions (Fig. 4B), where the best results were obtained with high ferredoxin and low ferredoxin reductase concentrations. Early literature reports have also highlighted the importance of retaining the natural *dnrV-doxA* genetic organization in expression studies[8]. Expression of *fdx4-dnrV-doxA* as a multigene operon using the strong synthetic *sp44* promoter resulted in a 2.8-fold increase in production of DXR to 166 mg/L. Interestingly, the addition of *fdr3* under the control of the moderately active *gapdh* promoter was detrimental to the formation of DXR, demonstrating that the correct stoichiometry of the redox partners was also relevant in vivo.

Cultivation of the engineered strains was conducted with LXA1180 resin to bind anthracyclines and alleviate toxicity issues. However, the use of the resin resulted in the appearance of an unexpected compound (Fig. 6C), which was identified as ε-rhodomycinone (Fig. 1) based on LC-MS and NMR analysis (Supplementary Table 6, Supplementary Fig. 16–S18). We optimized the culture conditions and found that the hydrophobic pathway intermediate did not accumulate in cultures enriched with XAD7 resin (Fig. 6C), where carbon flux to the end products DXR and DNR was further improved. Anthracycline production by the best performing strain G001/pSnorO +pFdx4_DnrV_DoxA was subsequently evaluated in a bioreactor in 20 L volume (Fig. 6D). Time-course analysis revealed production of 336 mg/L DXR with an 81:19 DXR:DNR ratio, a 180% increase in doxorubicin production, enhancing its biotechnological feasibility.

In summary, the anthracycline DXR was discovered in 1969 from cultures of *S. peucetius* ATCC27952 during industrial strain development[6]. However, despite decades of work and the importance of the compound for the pharmaceutical industry, DXR is still manufactured predominantly by semi-synthesis from DNR[1]. Here, we have uncovered several independent factors that limit the formation of DXR. First, we demonstrate that the activity of the P450 enzyme DoxA is highly dependent on the correct Fdx-FdR redox engine. Our transcriptomics data using industrial production strains uncovered *fdx4* as the preferred electron donor to DoxA. In turn, phylogenetic analysis and biochemical characterization revealed that the SpFdR3 ferredoxin reductase is the optimal natural reductant of SpFdx4. Future research focusing on the use of non-native or artificial redox partners may lead to further catalytic improvements[13], since wild-type *S. peucetius* does not naturally produce DXR and therefore evolution has not optimized the natural redox partners of DoxA. Second, we clarified the role of DnrV in the metabolic pathway and propose that it is an intracellular drug-binding protein, which may prevent the interaction of anthracyclines with DNA and subsequent toxic effects. The protein is indirectly vital for the biosynthesis of DXR due to the inherent product inhibition displayed by DoxA. However, the observation that DnrV also protects the anthracycline substrates from deglycosylation indicates a multifaceted role and warrants further investigations into the protein family. Third, the structure of DoxA in complex with DNR demonstrates that C13 of the substrate is positioned ideally for catalysis above the heme group. The preferred *anti*-conformation of DNR positions the C14 carbon in a small hydrophobic pocket pointing away from the heme unit. While other factors, such as electron transfer rate and the oxygen activation mechanism, have been noted to strongly influence P450 catalysis[33,36], the slow conversion rate to DXR suggests that

structure-based engineering of DoxA may be required to further improve catalysis. Despite these caveats, we were able to utilize the results presented here for rational metabolic engineering to generate a high-yield strain, paving the way for industrial manufacturing of biosynthetic DXR.

## Methods

### Genome assembly and annotation
Chromosomal DNA of the DNR and DXR overproducing mutants, G001 and X121, respectively, were extracted and their genomes were sequenced. The strains were cultivated (30 °C, 300 rpm) in 50 ml GYM medium consisting of glucose 4 g/L, yeast extract 4 g/L, malt extract 10 g/L, and supplemented with 0.5% glycine for 3 days. The cells were collected by centrifugation (3000 x *g*, 5 min). Genomic DNA was extracted using the method from Nikodinovic et al.[45] with minor modifications. The DNA was transferred to the Finnish Functional Genomics Centre (Turku, Finland) for PCR-free shotgun library preparation (Illumina) and sequenced using MiSeq v3, generating $2 \times 300$ bp paired-end reads (Illumina). The quality of the reads was manually checked before and after trimming and error correction using FASTQC (v0.11.2)[46]. Subsequently, the reads were error corrected and de novo assembled using A5-miseq (v20150522)[47]. Assembly of contiguous sequences was performed with ABACAS (v1.3.1)[48] using *S. peucetius* ATCC 27952 (CP022438.1) as the reference[49]. Gaps were filled using IMAGE (v2.4.1)[50]. The final assembly was annotated using RAST[51]. All programs were run with the default parameters on the CSC – IT Center for Science's Taito super-cluster (Espoo, Finland). The final assembly was deposited in the National Center for Biotechnology Information (NCBI) database under the accession numbers JBLNLL000000000 for G001 and JBLNLK000000000 for X121.

### Transcriptomics
Wild-type, G001 and X121 strains were cultivated (30 °C, 300 rpm) in quadruplicates in 50 ml of E1 medium (glucose 20 g/L, starch 20 g/L, farmamedia 5 g/L, yeast extract 2.5 g/L, $K_2HPO_4$ x 3 $H_2O$ 1.3 g/L, $MgSO_4$ x 7 $H_2O$ 1 g/L, NaCl 3 g/L, $CaCO_3$ 3 g/L). After cultivation, 1 ml of culture from each respective cultivation was pooled. Immediately after pipetting the samples into the tubes, 445 µl of STOP solution (95% EtOH, 5% phenol) was added. The tubes were vortexed for 5-10 s at full speed and centrifuged (4000 x *g*, 10 min). The supernatants were discarded, and the pellets were stored at -80 °C after freezing in liquid nitrogen. The cells were lysed with a mortar and pestle under liquid nitrogen. Total RNA was isolated using a RNeasy Mini Kit (Qiagen) with DNase treatment. Total RNA was sent to Finnish Functional Genomics Centre (Turku, Finland) for quality control (Agilent Bioanalyzer 2100), library preparation (Illumina), and sequencing with HiSeq 2500 (Illumina) to produce $1 \times 50$ bp reads. All analyses were performed using the Chipster platform[52]. The reads were manually checked using FASTQC[46], and trimming was performed using TRIMMOMATIC[53]. The trimmed reads were aligned to the genome using BowTie2[54] and counted using HTSeq[55]. Differential expression was performed using edgeR[56]. The RNA-Seq data have been deposited in GEO (GSE289319) for WT, G001, and X121, respectively.

### Phylogenetic analysis
The evolutionary history was inferred using the UPGMA method. The bootstrap consensus tree inferred from 888 replicates was taken to represent the evolutionary history of the taxa analyzed[57]. Branches corresponding to partitions reproduced in less than 50% bootstrap replicates are collapsed. The evolutionary distances were computed using the JTT matrix-based method[58] and are in the units of the number of amino acid substitutions per site. The rate variation among sites was modeled with a gamma distribution (shape parameter = 3). This analysis involved 15 amino acid sequences. All positions with less than 88% site coverage were eliminated, i.e., fewer than 12% alignment gaps,

missing data, and ambiguous bases were allowed at any position (partial deletion option). There were a total of 276 positions in the final dataset. Evolutionary analyses were conducted in MEGA11[59]. UniProt accession numbers for BphA4, PdR, SmFdR, AtFdR, MtFdR, ZmFdR, SoFdR, and PsFdR are Q52437, P16640, Q5S3I2, Q8W3L1, P9WIQ3, Q9SLP6, P00455, P10933, respectively.

## Protein production and purification

The proteins used in the in vitro assays were produced as N-terminally 6 × His-tagged recombinant proteins in *Escherichia coli* TOP10. Genes were ordered as synthetic DNA strands from either GeneWiz or Euro-fins Genomics, cloned into the plasmid vector pBADHisBΔ[60] with the restriction enzymes *Bgl*II and *Hind*III, and the expression construct was transformed into TOP10 by heat shock. Cells were grown in 330 ml of 2 × tryptone yeast extract medium (2xTY) until the cultures reached an $OD_{600}$ of 0.6–0.8, at which point protein production was induced by adding 0.02% L-arabinose. When producing iron-containing enzymes and DoxA, 0.1 mM of $(NH_4)_2Fe(SO_4)_2 \cdot 6H_2O$ and 0.25 mM aminole-vulinic acid, respectively, were added at the time of induction. Protein was produced overnight, and cells were collected by centrifugation, suspended in wash buffer (20 mM sodium phosphate buffer, pH 7.4, 150 mM NaCl, 5 mM imidazole, 10% (v/v) glycerol), and disrupted by sonication. Cell debris was removed by centrifugation, and TALON Superflow resin (GE Healthcare) was added to the crude extract. In the case of the reductases, 10 μM of FAD was also added to the crude extract during this step. After 1 h of incubation, the resin was moved to a column and washed with additional A buffer and then eluted with B buffer (20 mM sodium phosphate buffer, pH 7.4, 150 mM NaCl, 200 mM imidazole, 10% (v/v) glycerol). PD-10 columns (GE Healthcare) were used to change the buffer to buffer C (40 mM sodium phosphate buffer, pH 7.4, 300 mM NaCl, 10% (v/v) glycerol), and the protein was concentrated with Amicon Ultra-4 10 K centrifugal filters (Merck Millipore) down to 300–600 μl. Glycerol was added to 40% (v/v), and the proteins were stored at -20 °C. The purity and molecular weight of the proteins were checked by SDS-PAGE (Supplementary Fig. 2), and the concentration was estimated by the Bradford method[61]. Spinach (*Spinacia oleracea)* ferredoxin SoSfx and ferredoxin−NADP+ reductase SoFdR were obtained from Sigma.

For crystallization purposes, the *doxA* gene was cloned into the in-house vector pCPF3.11, a pET derivative containing an N-terminal enterokinase-cleavable His6-tag. The protein was expressed using BL21 (DE3) Rosetta2 cells by growing fresh transformants in 2xYT supplemented with Ampicillin (100 μg mL⁻¹) and chloramphenicol (30 μg mL⁻¹) to $OD_{600}$ 1.4 before inducing with 0.5 mM Isopropyl β-D-1-thiogalactopyranoside. Expression was performed at 25 °C for 3 hours and cells were harvested by centrifugation for 20 min at 4 kG, and the resulting pellets were stored at −20 °C. DoxA-expressing cell pellets were resuspended in buffer HisA (50 mM Tris pH 8.0, 300 mM NaCl, 20 mM imidazole) and lysed using sonication with these and all following steps conducted at 4 °C. The supernatant after centrifugation for 40 min at 30 kG was loaded on a 5 mL HisTrap FF crude column (Cytiva) preequilibrated in buffer HisA, and the DoxA protein was eluted using a gradient of buffers HisA and HisB (HisA with 500 mM Imidazole). Fractions containing DoxA were combined and diluted two-fold using buffer Q-A (50 mM Tris pH 8.0) before being loaded on an equilibrated HiTrap Q column (Cytiva) and eluted with a gradient of buffer B (Q-A with 1 M NaCl). DoxA-containing fractions were pooled and divided in 200 nmole aliquots before each was incubated with a substrate (if required) for 1 hr. The resulting mix was run on a Superdex75 16/60 (Cytiva) gel filtration column using 20 mM Tris pH 7.4 and 150 mM NaCl, with extra absorbance monitoring on 482 nm for anthracycline monitoring. Complex-containing fractions were analyzed on gel, pooled, and concentrated to at least 12 mg mL⁻¹ before being flash frozen or directly set up for crystallization.

## CYP450 CO-binding assay

The integrity of the heme group and the concentration of active DoxA were measured with a P450 CO-binding assay[62]. DoxA was diluted into a solution of 80 mM sodium phosphate buffer, 1 mM EDTA, and 10% glycerol, and the difference in absorption at 420 nm and 450 nm was measured before and after the addition of CO gas and sodium dithionite to the sample. The concentration was then calculated using the extinction coefficient 91,000 M⁻¹ cm⁻¹.

## Enzymatic assays

Enzymatic assays to measure DoxA activity were performed in vitro in reaction volumes of 200 μl. For analysis of FdR enzymes, DoxA (6 μM), ferredoxin (10 μM), ferredoxin reductases (10 μM), catalase (2.5 U), sodium phosphate buffer pH 7.5 (≈20 mM) and substrate (13-DOD, 13-DHD, DNR, ≈ 15 μM) were mixed with a NAD(P)H recycling system consisting of glucose (4 mM) and glucose dehydrogenase (0,5 U). For the commercially acquired spinach SoFdx and SoFdR, 8.8 μg and 0.02 U, respectively, were used in the reactions. For the reactions with PdR, only 0.5 μM reductase and an incubation time of 2 h were used, since higher concentrations led to degradation of the substrate. Optimized reaction conditions for full conversion to DXR included 10 μM DoxA, 20 μM SpFdx4, 0.1 μM SoFdR, 20 μM DnrV, 4 mM Glucose, 1U GDH, and 2,5 U catalase, 0.5 mM NADP+. The reactions were started via the addition of NAD(P)H and NAD(P)+ (0.5 mM) and left to react at room temperature overnight. The reactions were extracted with a 4:1 mix-ture of chloroform and methanol, dried *in vacuo*, dissolved in pure methanol, and analyzed by HPLC (SCL-10Avp/SpdM10Avp system with a diode array detector (Shimadzu)) using a C18 Kinetex column (2.6 μM, 100 Å, 4.6 × 100 mm (Phenomenex)) at room temperature. The gradient run parameters included 0-2 min, 0% B; 2−20 min, 0−60% B; 20-24 min, 100% B; 24−29 min, 0% B. Solvent A consisted of 15% acetonitrile in 0.1% HCOOH, and solvent B of 100% acetonitrile. The absorbance of the samples was recorded at 490 nm.

## Crystallization of DoxA

All DoxA crystals were grown at 20 °C by sitting drop vapor diffusion by mixing mother liquor and protein solution in a 1:1 volumetric ratio. Crystals of DoxA in complex with DNR were grown at 20 °C by sitting drop vapor diffusion in Intelli-Plate 96-2 shallow well plates (Hampton Research) by combining 1 μL of protein solution containing 12 mg mL⁻¹ DoxA and 1 mM DNR with 1 μL of well solution composed of 1.65 M ammonium sulfate, 0.1 M Bis-Tris pH 5.5, 2% trehalose. The droplets were seeded immediately by streaking with a cat whisker charged with previously grown DoxA crystals. The crystals were cryoprotected by the direct addition of 8 μL of a cryoprotectant solution composed of 1.65 M ammonium sulfate, 0.1 M Bis-Tris pH 5.5, 18% trehalose, 10 mM Hepes pH 7.5, 50 mM NaCl, 2% glycerol, 1 mM DNR, and flash frozen in liquid nitrogen. Diffraction data was collected at the University of Michigan Center for Structural Biology using an Excillum MetalJet-D2 + 70 kV, 250 W X-ray source equipped with an Eiger2 R 4 M hybrid detector. The data were integrated and scaled using XDS[63] and the structure was solved by molecular replacement using Phaser MR[64] with CYP154C2 (pdb: 7CL9) as the search model. The initial model was subjected to iterative rounds of manual building and refinement using Coot[65] and Phenix.refine[66]. For the crystals of complexes with DOD and DHD, DoxA was pre-incubated for 1 hr with the substrate before being subjected to gel filtration. Peak fractions were concentrated to 16 mg /mL and set up in MRC 2-well plates using an NT8 robotic crystal screen dispenser (Formulatrix) on a 200 nL scale. Crystals for DoxA-DOD were found in conditions containing 0.1 M citrate pH 5.0 and 20% (w/v) PEG6000, whilst for DHD-complex a mother liquor solution with 0.1 M MES pH 6.0, 0.2 M NH₄Cl and 20% (w/v) PEG6000 yielded diffracting crystals. Single crystals were cryoprotected with 30% ethylene glycol before being flash-frozen. Diffraction data were collected at the European Synchrotron Radiation Facility, in particular the MASSIF

beamlines, and processed using DIALS[67]. Data were loaded in the CCP4[68] pipeline: data were reduced using Aimless[69] and the structure was solved using PHASER[64] by molecular replacement with the DoxA AlphaFold model (AF-A0A2D3UDU4). The resulting model was finalized using iterative rounds of refinement with REFMAC[70] and model building in COOT[65].

## Quantum mechanics

All QM calculations were performed with Gaussian 16[71]. Geometry optimizations and frequency calculations were performed on the proposed crystal structure conformations at the B3LYP level with 6-31G(d). Optimized structures contained zero negative frequencies.

## Molecular dynamics simulations

The heme iron (IV)-oxo complex involved in the cytochrome-catalyzed oxidation (compound I) was used to model the active form of the heme cofactor. Simulations were performed using the Amber 12 package[72], and the Amber-compatible parameters[73] were used for compound I and its axial Cys ligand. Parameters for anthracycline ligands (DOD and DNR) were generated within the antechamber module in AMBER force field (gaff), with partial charges set to fit the electrostatic potential generated at the HF/6-31 G(d) level by the RESP model[74]. The charges were calculated according to the Merz-Singh-Kollman scheme[75,76] using Gaussian 16. DoxA was immersed in a pre-equilibrated truncated cuboid box with a 10 Å buffer of TIP3P[77] water molecules using the tleap module. All subsequent calculations were done in Amber/ff99SBildn force field[78]. We then performed 1000 ns simulations starting from the crystal structure of DoxA bound to deoxy-daunorubicin (DOD) and daunorubicin (DNR). The system was optimized, gradually heated to 300 K, and equilibrated before 1000 ns simulation runs. Simulations were analyzed using Chimera[79], and figures were made using PyMol (The PyMOL Molecular Graphics System, Schrödinger, LLC).

## Generation of sufR mutant strain

The disruption of the *sufR* gene was carried out via homologous recombination using the unstable multicopy *pWHM3*-based vector[80], *pWHM3-oriT*. The disruption cassette contained the apramycin resistance gene, *aac(3)IV*, flanked by 1 kb regions upstream and downstream of *sufR*. This cassette was synthesized as synthetic DNA and cloned into *Hind*III and *Xba*I restriction sites of the multicopy *pWHM3-oriT* vector. The resulting construct was introduced into *E. coli* ET12567/*pUZ8OO2* strain[80] and conjugated with *S. peucetius* ATCC27952 strain. Mutants were generated through several passages of exconjugants under non-selective conditions on MS plates. Double-crossover mutants were screened based on apramycin resistance and the loss of thiostrepton resistance. Finally, PCR and gel electrophoresis confirmed the deletion of the target gene (Supplementary Table 7, Supplementary Fig. 1).

## Construction of plasmids and strains

The *doxA*, *dnrV*, *fdx4*, and *fdr3* genes from *S. peucetius* ATCC 27952 and *snorO* gene from *S. nogalater* ATCC 27451[37], along with their corresponding ribosome binding sites, were ordered as synthetic fragments from GeneWiz. The *dnrV-doxA* sequence was synthesized as fragments from the genome while keeping their native sequence intact. The integrative vectors pSET152[40], pSET154BB[81], and pEN-SV1[41] were used to express biosynthetic genes. The *E. coli* TOP10 strain was utilized for cloning and plasmid propagation. DNA fragments were assembled with vectors using corresponding restriction sites. The *snorO* gene, flanked with EcoRI/PstI restriction sites, was cloned into the pEN-SV1 vector. Then, for the first round of constructs, the *doxA* gene, flanked with *Xba*I/*Pst*I, was added downstream of the *snorO* gene in the pEN-SV1 vector. The pSET152 vector was constructed to harbor the *fdx4* and *fdr3* genes, flanked with *Eco*RI/*Pst*I and *Spe*I/*Pst*I, respectively, followed

by the introduction of the *dnrV* gene with *Pst*I/*Nsi*I. For the second round of constructs, *fdx4* gene, the *dnrV-doxA* fragment and *fdr3* gene were assembled using BioBricks and cloned in pSET154BB with *Eco*RI/*Pst*I. The fd phage terminator[82] was introduced after *dnrV-doxA* fragment. Promoters and RBSs are indicated in Fig. 6. The final constructs were integrated into the genome of *S. peucetius* G001 with conjugation using standard protocol. The *E. coli* ET12567/pUZ8002 strain was used for conjugation.

## Culture conditions and metabolite extraction

For comparison of WT, G001, and X121, strains were cultivated in triplicate in 50 ml of E1 medium for 13 days. Anthracyclines aglycones were extracted using acid hydrolysis. Specifically, 2 mL of culture samples were collected, mixed with 2 mL of 1 M HCl, and boiled for 20 minutes. After cooling, 2 mL of chloroform was added, and the samples were mixed for 30 minutes. The lower phase was then dried in a concentrator and dissolved in 200 μL of methanol before HPLC analysis. The *S. peucetius* G001 strain and its mutants were cultivated in triplicate in 30 mL of E1 medium with the addition of 20 g/L of either LXA1180 (Sunresin) or XAD7 HP (Supelco) resin. After 5 days, the resin was collected, washed, and the produced compounds were extracted with 25 mL of methanol. For the best doxorubicin-producing mutant, 1 mL of culture was collected at various time points, the resin was washed from the cells, and the produced compounds were extracted with 1 mL of methanol. The HPLC analysis of samples was performed at room temperature using a Shimadzu Nexera X3 system with a photo-diode array detector and a C18 column (2.6 μM, 100 Å, 4.6 × 100 mm Kinetex column (Phenomenex)). A total of 10 μL was injected. The gradient run parameters included 0-2 min, 0% B; 2–20 min, 0–60% B; 20-24 min, 100% B; 24–29 min, 0% B. Solvent A consisted of 15% acetonitrile in 0.1% HCOOH, and solvent B of 100% acetonitrile. Absorbance was measured at 490 nm, and the reaction products were identified by comparison to standard compounds. The yield of compounds was calculated using doxorubicin and daunorubicin standard curves.

For large-scale production, strain G001/pSnorO +pFdx4_DnrV_DoxA was cultivated in a 42 L total volume Bioengineering LP bioreactor (Bioengineering AG). Seed cultures were prepared in 250 mL flasks using 50 mL of E1 medium and incubated at 30 °C and shaking at 250 rpm. A 10% (v/v) inoculum was transferred into the bioreactor containing 20 L of E1 medium supplemented with XAD-7 resin (20 g/L) at the start of fermentation. The fermentation was carried out for 8 days at 30 °C with an airflow rate of 15 NL/min. The dissolved oxygen (DO) setpoint was maintained at 40% by controlling agitation in a cascade mode, from a baseline of 250 rpm up to 300 rpm. Polypropylene glycol 2000 was added as an antifoam agent as needed. Samples were collected every 24 hours for compound extraction and subsequent HPLC analysis as previously described.

## Purification and identification of compounds

For the structure elucidation of ε-rhodomycinone, methanol extracts from LXA1180 resin were purified by silica chromatography using high-purity grade silica (pore size 60 Å, 230-400 mesh particle size) and 250 mL of an eluent at 49.5:49.5:1 MeOH/DCM/HCOOH and 100 mL of an eluent at 59.4:39.6:1 MeOH/DCM/HCOOH. Fractions from silica chromatography were further purified by semi-preparative HPLC using a LC-20AP/CBM-20A system with a diode array detector (Shimadzu) and Phenyl-Hexyl, 5 μm, 100 Å, 250 × 21.2 mm Kinetex column (Phenomenex) The gradient run parameters included 0–2 min, 0% B; 2–22 min, 0–100% B; 22–24 min, 100% B with a flow rate of 20 mL/min. Solvent A consisted of 15% acetonitrile in 0.1% HCOOH, and solvent B of 100% acetonitrile. Fractions containing ε-rhodomycinone were extracted with chloroform, dried using a rotary evaporator and desiccator, and resuspended in deuterated solvents for NMR measurements. High-resolution electrospray ionization mass spectra were

recorded by Waters ACQUITY RDa Detector using an XBridge BEH C18 Column, 130 Å, 5 μm, 4.6 × 30 mm (Waters). NMR spectra were recorded by a 500 MHz Bruker AVANCE-III system with a liquid nitrogen-cooled Prodigy BBO cryoprobe. All NMR spectra were processed in Bruker TopSpin 3.7.0 version, and the signals were internally referenced to the solvent signals.

For the structure elucidation of 7-deoxydaunorubicinone, 10.5 mg of daunorubicin was subjected to biocatalytic reduction by spinach ferredoxin reductase SoFdR. The reaction was performed overnight at 28 °C in a degassed buffer with the following components: 2 mM daunorubicin, 1 mM NADP+, 5 mM glucose-6-phosphate, 1 U/μL glucose-6-phosphate dehydrogenase, 10 μM spinach ferredoxin reductase, 1% DMSO, 10% glycerol, 50 mM potassium phosphate, pH 7.4. Precipitation of the aglycone was observed after incubation. The reaction was extracted four times with an equal volume (10 mL) of chloroform. The first solvent addition produced an emulsion that was treated by bath sonication and centrifugation at 1000 x $g$ for 5 min. The combined organic layers were washed twice with an equal volume of brine, dried over $Na_2SO_4$, and concentrated by rotary evaporation to afford 7-deoxydaunorubicinone in quantitative yield. HR-MS analysis was performed on Agilent 6545 Q-TOF. NMR spectra were recorded by Varian 600 MHz NMR with nitrogen cooled Prodigy cryoprobe. All spectra were processed with MNova software.

### Statistics and reproducibility
No statistical method was used to predetermine sample size. Sample sizes were chosen based on standard practice in the field and on the number of independent biological replicates required to ensure reproducibility. No data were excluded from the analysis. Bacterial cultivations for metabolite production analysis were performed in independent biological triplicates (n = 3) unless otherwise stated in the text. Results of experiments were presented as mean ± standard deviation (SD), as indicated in the legends of the figures. Molecular dynamics simulations were performed in triplicate (n = 3). Differential expression analysis for transcriptomics was performed using the statistical methods implemented in edgeR.

### Reporting summary
Further information on research design is available in the Nature Portfolio Reporting Summary linked to this article.

## Data availability
All data supporting the findings of this study are available within the Article and Supplementary Information file. The final genome assembly data were deposited in the National Center for Biotechnology Information (NCBI) database under the accession numbers JBLNLL000000000 for G001 and JBLNLK000000000 for X121. The RNA-Seq data have been deposited in the Gene Expression Omnibus (GEO) database under the accession number GSE289319. The atomic coordinates and structure factors generated in this study have been deposited in the Protein Data Bank. The DoxA–DHD, DoxA–DOD, and DoxA–DNR structures are available under accession codes 9SI5, 9S7F and 9O35, respectively. Source data are provided with this paper.

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

## Acknowledgements

Support of this research from the Novo Nordisk Foundation NNF19OC0057511 (to M.M.-K.), the Academy of Finland Grants 340013 and 354998 (to M.M.-K.), NIH grant R35 GM118101 (to D.H.S.), Hans W. Vahlteich Professorship (to D.H.S.), NSF Grants CHE-2153972 (to K.N.H.), ENG-2321976 and CHE-2348596 (to S.E.N.), and the UM Pharmacological Sciences Training Program (H.B) is gratefully acknowledged. We thank Dr. Tero Kunnari (Heraeus GmbH) for the gift of anthracycline reference samples and Dr. Kristiina Ylihonko (Care4living Oy) for *Streptomyces* strains, whilst Dr. Dennis Wander is acknowledged for supplying synthetic DOD. The authors would like to thank Patrick Voskamp (Leiden University crystallization facility) and Matthew Bowler and other beamline scientists at ESRF (MASSIF-1 and -3) for their support in crystallography experiments.

## Author contributions

A.K., E.A., K.Y., and S.A.N. contributed equally to this work. M.K. and K.Y. designed and performed the experiments related to microbiology and transcriptomics. A.K. contributed to fermentation and metabolic engineering experiments. H.T. and M.B. contributed to compound purification and structure elucidation. E.L. performed the large-scale fermentations. H.B., J.S., and R.J.H. carried out the computational modelling and DFT/MD analyses. S.A.N., R.Q.K., M.H., and R.C.M.K. performed protein purification, crystallography, and structural analysis. R.W. contributed to strain construction. E.A. and M.L. carried out protein purification and in vivo assays. M.M.-K., J.N., G.P.W., J.J.N., S.E.N., D.H.S., and K.N.H. conceptualized and supervised the project. All authors contributed to the revision and editing of the final written draft.

## Competing interests

The authors declare no competing interests.
