## [Peer Review File · Nature Communications]

Metabolic Engineering of Doxorubicin Biosynthesis through P450-Redox Partner Optimization and Structural Analysis of DoxA

Corresponding Author: Professor Mikko Metsä-Ketelä

Version 0:

Reviewer comments:

Reviewer #1

(Remarks to the Author)

In this paper, the natural electron transfer chaperone FDX4-FDR3 was systematically identified and validated for the first time by comparing transcriptomic data, which significantly enhanced the catalytic efficiency of DoxA and solved the key problem of the P450 enzyme, which has long been an “unknown source of electrons”. Subsequently, the authors revealed that DnrV, as the first drug-binding protein in the VOC superfamily that interacts with P450, indirectly promotes DoxA catalysis by preventing product inhibition through binding to DXR, thus expanding the understanding of P450 enzyme regulation. By resolving the crystal structure of DoxA complexed with three substrates, combined with molecular simulations, the conformational mechanism (anti-conformation advantage) of DNR being difficult to be hydroxylated was elucidated, which provides a theoretical basis for the subsequent structural modification. Based on the above discoveries, the authors have constructed an engineered strain with high yield of DXR, which can increase the yield by 180%, and has the value of practical industrialization application. The paper covers multi-dimensional means, such as histological analysis, protein expression purification, in vitro enzymology, structural biology, computational chemistry and metabolic engineering, etc., which are cross-validated, logical, and highly convincing. However, the manuscript needs to be improved for publication in Nature Communications. Some comments should be addressed below.

Major points:

The exploration of the dynamics of the crystal structure of DoxA is still insufficient, although the authors have performed 1000 ns molecular dynamics simulations, and it is recommended to supplement the statistical analysis of conformational changes induced by different substrates (e.g., changes in the volume of the active pocket, RMSF of key residues, etc.) to explain the differences in substrate selectivity in a more specific way.

The article focuses on the role of FDX4/FDR3, but no comparative experiments with other potential FDX/FDR combinations (e.g., fdx5 mentioned in the article) were performed, which may affect the generalizability of the conclusions. It is recommended to supplement the in vitro activity comparison of different FDX/FDR combinations to verify whether FDX4/FDR3 is the optimal choice.

The molecular mechanisms (e.g., binding sites, dynamic interactions) of DnrV binding to DXR have not been validated by co-crystallization or mutagenesis experiments. It is recommended to utilize targeted mutagenesis or structural biology to clarify the key residues of DnrV binding to DXR and to verify its regulatory role on product inhibition.

The percentage of DNR “productive poses” in the molecular dynamics simulation was only 14%, which needs to be further statistically verified for significance. It is recommended to increase the number of simulation repetitions or adopt enhanced sampling techniques (e.g. Metadynamics) to improve the reliability of conformational sampling.

Data on long-term passaging stability of engineered strains, scale-up of fermentation (e.g., bioreactor level), and economic analysis (e.g., carbon source utilization efficiency) are lacking. It is recommended to supplement the results of continuous

culture experiments and pilot-scale fermentation to assess its industrialization feasibility.

Some of the experimental steps are described briefly (e.g. DoxA purification conditions, crystallization conditions optimization), which may affect the reproducibility. It is suggested to add key parameters (e.g., buffer pH, temperature, protein concentration) and quality control data (e.g., standard curve for enzyme activity assay).

Other potential factors (e.g., electron transfer rate, oxygen activation mechanism) for the inefficiency of DoxA-catalyzed DNR to DXR were not adequately discussed. It is recommended to include in the discussion an exploration of possible rate-limiting steps and to suggest future research directions (e.g. cofactor engineering or dynamic regulatory strategies).

Graphical and Data Presentation: Error calculation methods: Specify the type of error bars (standard deviation SD or standard error SEM) in the legend or methods section, and indicate the number of replicates (e.g., "n=3 biological replicates"). If there are batch-to-batch differences, supplement the data points of a single experiment with a boxplot or scatterplot. Detailed description of the HPLC gradient program (e.g., "0-2 min, 0% B; 2-20 min, 0-60% B") with mobile phase composition (e.g., "A: 0.1% formic acid aqueous solution, B: acetonitrile") in the Methods section. Add the detection wavelength (e.g., " $\lambda = 490 \text{ nm}$ ") and column temperature (e.g., " 25°C ").

Reviewer #2

(Remarks to the Author)

• What are the noteworthy results?

1. The natural redox partners ferredoxin FDX4 and ferredoxin reductase FDR3 were identified through transcriptomic analysis, which are crucial for DoxA catalysis.

2. Structural analysis and density functional theory (DFT) calculations revealed that the inefficient C-14 hydroxylation of daunorubicin is due to the unfavorable anti-conformation of its methyl ketone sidechain.

3. Advances in rational strain engineering led to a 180% increase in doxorubicin yields, improving the production profile.

• Will the work be of significance to the field and related fields? How does it compare to the established literature? If the work is not original, please provide relevant references.

There is some progress, but it is not significant. The main achievement is obtaining the three-dimensional structures of the complexes formed by the DoxA enzyme and the substrates DNR, DHD, and DOD.

• Does the work support the conclusions and claims, or is additional evidence needed?

Additional evidences are needed. For example, high-resolution three-dimensional structure of apo DoxA, and direct use of DNR as substrate to verify the product inhibition of DXR, etc.

• Are there any flaws in the data analysis, interpretation and conclusions? Do these prohibit publication or require revision? Yes, I have attached a file. I believe these should prohibit publication.

• Is the methodology sound? Does the work meet the expected standards in your field?

No, I do not believe the methodology is sound, and the work does not meet the expected standards in the field.

• Is there enough detail provided in the methods for the work to be reproduced?

The substrate DOD is not readily available. Additionally, reproducing the other data as a whole is relatively straightforward.

Reviewer #3

(Remarks to the Author)

This is an excellent study on engineering a P450 system for improved production of doxorubicin, an important chemotherapeutic agent. The authors have improved catalysis by finding the natural redox partners for DoxA and by using an anthracycline binding protein to block product inhibition of the P450. I have only a few minor comments.

1. Somewhere in the paper the author should give the correct CYP name for DoxA.

2. In the figure legend to figure 4 they refer to panel "f" as "e".

3. The MD doesn't add much especially using compound I. The lifetime of compound I is on the order of picoseconds, as estimated from radical clocks, so a 1000 ns simulation really is not of much relevance. A better model would be the oxy-complex. There are a few good P450-oxy-complex structures as a basis for modeling.

4. In the MD word they only report the percentage of frames where C14 is in position for hydroxylation. They should also report the percentage of frames for C13.

Version 1:

Reviewer comments:

Reviewer #2

(Remarks to the Author)

Agree to proceed with the publication of the article.

Reviewer #3

(Remarks to the Author)

The authors have addressed my main comments regarding molecular dynamics. They've included more runs which gives better statistics. They still argue that long simulations of compound I are justified because everyone does it this way. That may be so, but it doesn't mean it's correct thing to do. In the long run it would only be significant if the substrate wandered far away over the long simulation but that's not the case here so everything is okay.

- Is the CC (1/2) value for DoxA *apo* reasonable? It is significantly lower than 0.5. The Rpim value is 0.902, which is much greater than 0.5. The resolution is too low, resulting in lower reliability for the overall *apo* structure.
- Is "Na₂SO₄" written incorrectly in Figure S3? Additionally, is the formatting consistent with academic standards?
- The compound 7-deoxydaunoribicinone does not appear anywhere in the main text of the manuscript.
- Line 67: V_{max} values writing specification?
- Line 143-145:

The manuscript states that "FDX4 is a 2Fe-2S ferredoxin similarly to SFX and PDR," but later concludes that FDX4 is a Rieske-type ferredoxin while PDR is classified as a non-Rieske [2Fe-2S] ferredoxin (Figure S3). This raises two concerns:

1. **Logical inconsistency:** If FDX4 shares similarity with PDR, their classifications as Rieske-type and non-Rieske ferredoxins, respectively, appear contradictory.

2. **Functional misclassification:** The description of PDR as a ferredoxin may require re-evaluation. Ferredoxin reductases (e.g., PDR family proteins) and ferredoxins (e.g., SFX family) are functionally distinct redox partners, not homologous proteins. This mischaracterization undermines the comparative analysis.

- Line 149-150:

The statement that "Five of the six FDR proteins related to PDR were produced in *E. coli*" (R1, R2, R3, R4, R7) raises methodological concerns:

1. **Unclear selection criteria:**

The manuscript does not define which six FDR proteins are considered "related to PDR" or specify the criteria for their selection. The phylogenetic tree (Figure 3B) does not clearly demarcate a clade containing exactly six FDR proteins. For instance, the purple-clade region (putative PDR-related group) appears to contain more than six members.

The rationale for excluding one protein from heterologous expression requires justification. Was this due to failed expression/purification, or evolutionary divergence?

2. **Phylogenetic inconsistencies:**

The claimed PDR-related FDRs (R1, R4, R7) cluster within a subclade distinct from other FDRs, yet R2 and R3 appear evolutionarily distant from this group. This contradicts the implied functional relationship.

If evolutionary proximity to PDR is the basis for selection, the six candidates should be explicitly annotated on the phylogenetic tree with bootstrap support values.

- Enzyme activity assay section: In Fig. 4A, using DOD as the

substrate, only a small amount of DXR is produced at a DnrV concentration of 10 μM , while at 20 μM , the yield of DXR is somewhat higher, but still lower than that of DNR. However, in Fig. 4C, when examining the effect of DXR concentration on the reaction, DOD is again used as the substrate, and at a DnrV concentration of 10 μM , the amount of DXR produced in the absence of added DXR is significantly higher than in Fig. 4A, even exceeding the yield of DNR. This presents a contradiction, as indicated by the red box in the figure below.

- Line 168-170:

Why was DOD used as the substrate in the experiment examining the effect of DXR concentration on the reaction, rather than DHD or DNR?

How can we determine whether the inhibition of the DoA reaction is due to the DXR product or if DoxA exhibits catalytic region selectivity?

- Line 400: 16 mg mL⁻¹. Pay attention to format specifications.

RESPONSES TO REVIEWER COMMENTS:

Reviewer #1 (Remarks to the Author):

In this paper, the natural electron transfer chaperone FDX4-FDR3 was systematically identified and validated for the first time by comparing transcriptomic data, which significantly enhanced the catalytic efficiency of DoxA and solved the key problem of the P450 enzyme, which has long been an “unknown source of electrons”. Subsequently, the authors revealed that DnrV, as the first drug-binding protein in the VOC superfamily that interacts with P450, indirectly promotes DoxA catalysis by preventing product inhibition through binding to DXR, thus expanding the understanding of P450 enzyme regulation. By resolving the crystal structure of DoxA complexed with three substrates, combined with molecular simulations, the conformational mechanism (anti-conformation advantage) of DNR being difficult to be hydroxylated was elucidated, which provides a theoretical basis for the subsequent structural modification. Based on the above discoveries, the authors have constructed an engineered strain with high yield of DXR, which can increase the yield by 180%, and has the value of practical industrialization application. The paper covers multi-dimensional means, such as histological analysis, protein expression purification, in vitro enzymology, structural biology, computational chemistry and metabolic engineering, etc., which are cross-validated, logical, and highly convincing. However, the manuscript needs to be improved for publication in Nature Communications. Some comments should be addressed below.

RESPONSE: Thank you for the overall evaluation.

Major points:

The exploration of the dynamics of the crystal structure of DoxA is still insufficient, although the authors have performed 1000 ns molecular dynamics simulations, and it is recommended to supplement the statistical analysis of conformational changes induced by different substrates (e.g., changes in the volume of the active pocket, RMSF of key residues, etc.) to explain the differences in substrate selectivity in a more specific way.

RESPONSE: We have analyzed the major hydrogen bond contacts between DoxA and either DOD or DAU through 1 μ s simulations run in triplicate, and this data is summarized in Figure S15. We note the differences between DOD and DAU, specifically that of DAU with Thr265 in line “240-242” of the main text.

The article focuses on the role of FDX4/FDR3, but no comparative experiments with other potential FDX/FDR combinations (e.g., *fdx5* mentioned in the article) were performed, which may affect the generalizability of the conclusions. It is recommended to supplement the in vitro activity comparison of different FDX/FDR combinations to verify whether FDX4/FDR3 is the optimal choice.

RESPONSE: Thank you for the suggestions. As explained in more detail below, we were not able to test all FDX/FDR combinations as many of the proteins were not soluble. In the case of ferredoxins, our transcriptomics data revealed overexpression of only two genes *fdx4* and *fdx5*. We expressed both of these genes in *E. coli* for production of recombinant proteins, but *Fdx5* could not be produced in soluble form. Based on sequence analysis, *Fdx5* contains a [Fe4-S4] cluster that is likely to be highly sensitive to molecular oxygen, but protein production was not improved even under anoxic conditions. In the case of ferredoxin reductases, the transcriptomic data did not identify any strong ferredoxin reductase candidates likely to function with *Fdx4* and DoxA. We did clone all seven *fdr* genes that were identified from the genome of *S. peucetius* but were able to only produce five enzymes in soluble form in *E. coli*. As also noted by Reviewer #2, this was stated too briefly in the original submission, and we have re-phrased the text in line 159. For

these reasons, we believe that we have completed the in vitro activity comparison with all protein pairs that could be feasibly tested and the data is presented in Figure 3B.

The molecular mechanisms (e.g., binding sites, dynamic interactions) of DnrV binding to DXR have not been validated by co-crystallization or mutagenesis experiments. It is recommended to utilize targeted mutagenesis or structural biology to clarify the key residues of DnrV binding to DXR and to verify its regulatory role on product inhibition.

RESPONSE: We agree that the understanding of the molecular mechanism of DnrV is highly important and the topic has been the target of our investigations for several years. The research has expanded beyond anthracycline biosynthesis and has uncovered that DnrV is a member of a novel subgroup of VOC enzyme whose genes are often found co-located with P450 partner genes in numerous metabolic pathways. The common feature appears to be that the P450/VOC enzyme combinations are found from pathways where the substrates contain quinone units, such as in the biosynthesis of the vitamin K precursor menadione. The studies have expanded to such a degree that we feel that inclusions of structural studies of DnrV in this submission is challenging. A preprint of the DnrV work containing the three-dimensional structure, structure/function studies, more detailed biochemical characterization and enzyme kinetics is available at:

<https://chemrxiv.org/engage/chemrxiv/article-details/688a543623be8e43d6e90942>

However, we do recognize the need to reinforce our findings on DnrV in this work. Thus, we have expanded our analysis of the role of DnrV in the revised manuscript as follows. In the original submission, we only briefly mentioned the formation of deglycosylation shunt products. This is a previously known phenomenon that occurs even in the absence of P450 enzymes through direct reduction of anthracycline quinones by, for instance, spinach ferredoxin reductase (Fisher et al, 1985)). In the revision, we now present a new control reaction depicting formation of deglycosylation products under suboptimal reaction conditions under excess Pdx/PdR (lines 142 and Figure S4). This was done to highlight the role of DnrV in the protection of anthracycline substrates from degradation (revised Figure 4B). We have expanded the discussion on this point (line 173) and confirmed that substrate protection is specific for DnrV. As a negative control, we demonstrate that the related VOC enzyme, the Mitomycin Resistance Determinant (MRD), does not bind anthracyclines and therefore the reaction leads to formation of deglycosylated shunt products discussed in line 191 and shown in a new Figure S12.

Overall, we conclude that DoxA and improving doxorubicin production should be the focus of the current manuscript, while the focus of our follow-up manuscript will be on DnrV and the role of VOC enzymes more broadly on P450 catalysis.

The percentage of DNR “productive poses” in the molecular dynamics simulation was only 14%, which needs to be further statistically verified for significance. It is recommended to increase the number of simulation repetitions or adopt enhanced sampling techniques (e.g. Metadynamics) to improve the reliability of conformational sampling.

RESPONSE: Thank you for this suggestion, we have repeated our MD simulations in triplicate and reported these values in a new Table S5.

Data on long-term passaging stability of engineered strains, scale-up of fermentation (e.g., bioreactor level), and economic analysis (e.g., carbon source utilization efficiency) are lacking. It is recommended to

supplement the results of continuous culture experiments and pilot-scale fermentation to assess its industrialization feasibility.

RESPONSE: We agree that demonstrating the potential of our engineering efforts for industrial DXR manufacturing is important. In this revision, we have scaled the production of DXR to a 20L volume in a bioreactor, which retained the overproduction of DXR over DNR (81:19 ratio) with slightly increased yields 336 mg / L DXR. This information has been added to the revised manuscript (lines 274) and replaces the time-course analysis of DXR production that was previously conducted in flask cultures (Figure 6D).

In terms of strain stability, we performed seven rounds of continuous cultivations in 250 mL flasks to observe product formation. Each flask was inoculated on day 3 from the previous generation and product formation was monitored on day 7. Based on the figure below, we observed increased DXR production in our engineering strain (left) throughout the experiments over the parental control strain (right). However, our data also demonstrated that anthracycline yields decreased after each cultivation round. Thus, our conclusion from this data is that the overproducing strain G001 is not suitable for industrial production, but that our P450 redox engineering appears to be stable as demonstrated by the DXR/DNR production profiles. We have decided not to include these data to the manuscript due to the loss of productivity.

In terms of economic analysis, we were severely hampered by lack of public information on industrial strain yields and costs of semi-synthesis of DXR from DNR, which are highly guarded trade secrets. Our consultation with industry representatives suggested that the semi-synthesis consists of five steps and synthesis yields of 50% are considered sufficient. In terms of carbon utilization, the fermentations were performed in an industrial medium that was utilized for anthracycline manufacturing in the early 2000s, but that product yields of 1-2 g / L of DNR are required for commercialization. When DNR to DXR semi-synthesis yields of 50% are taken into consideration, our engineering strains appear to be slightly below the target value. We would like to highlight that in our view this does not diminish the value of our P450 redox engineering, since transfer of our expression system to a contemporary industrial host strain should retain high DXR:DNR ratio with improved yields. However, we have not been able to acquire current industrial production strains for analysis. We have not added any discussion on the economic analysis to the manuscript, since we feel that unofficial discussions with industry representatives do not meet the rigor required for scientific publication.

Some of the experimental steps are described briefly (e.g. DoxA purification conditions, crystallization conditions optimization), which may affect the reproducibility. It is suggested to add key parameters (e.g., buffer pH, temperature, protein concentration) and quality control data (e.g., standard curve for enzyme activity assay).

RESPONSE: Agreed and amended. We have improved the Methods section and added details for DoxA purification and crystallization (starting at line 411).

We also agree that addition of quality control data for enzymatic activity (standard curves mentioned above, boxplots or scatterplots mentioned below) would improve the rigor of the work. However, generating this data has proven to be highly challenging, and time-demanding. The key issue is that optimization of the multienzyme reaction conditions is the result of years of work. The challenge has been that formation of deglycosylation shunt products occurs readily if the Fdx-FdR-DoxA-substrate ratio is not optimized. This key point is discussed in the revised manuscript at line 140 and displayed in Figure S4. We have identified the following factors that influence product yields and formation of shunt products:

- We have noted that each Fdx-FdR-DoxA combination requires individual optimization for best performance
- We have observed activity loss in storage for DoxA and Fdx4
- Heterologous expression of Fdx4 in *E. coli* has given variable yields of holoprotein generating batch-to-batch variation
- We have observed loss in activity with FdR with repeated freeze thaw cycles.
- We have found that inclusion of catalase is required, even with DnrV, indicating that some hydrogen peroxide is formed even under optimal reaction conditions

These issues have prevented us from presenting further quality control data. We would like to highlight that due to the challenges in the enzymatic assays, all data presented in any single figure in the manuscript always originate from parallel enzymatic reactions that were conducted at the same time on the same day. We deemed that this approach was necessary to prevent influence from batch-to-batch variation and storage stability issues. Reaction outcome data presented in the revision are based on optimized conditions providing the highest conversion rates.

Other potential factors (e.g., electron transfer rate, oxygen activation mechanism) for the inefficiency of DoxA-catalyzed DNR to DXR were not adequately discussed. It is recommended to include in the discussion an exploration of possible rate-limiting steps and to suggest future research directions (e.g. cofactor engineering or dynamic regulatory strategies).

RESPONSE: Agreed and amended. Discussion on the implications of our findings was limited in our original submission due to length restraints (<3500 words). However, we fully agree that it is important to address these issues in the Conclusions section. Thus, we have briefly outlined possible future research directions for each of the three major findings (DoxA redox engine; role of DnrV; structure of DoxA) in the revised manuscript.

Graphical and Data Presentation: Error calculation methods: Specify the type of error bars (standard deviation SD or standard error SEM) in the legend or methods section, and indicate the number of replicates (e.g., “n=3 biological replicates”). If there are batch-to-batch differences, supplement the data points of a single experiment with a boxplot or scatterplot. Detailed description of the HPLC gradient program (e.g., “0-2 min, 0% B; 2-20 min, 0-60% B”) with mobile phase composition (e.g., “A: 0.1% formic acid aqueous solution, B: acetonitrile”) in the Methods section. Add the detection wavelength (e.g., “ $\lambda = 490 \text{ nm}$ ”) and column temperature (e.g., “25°C”).

RESPONSE: Agreed and amended. We have added error bar descriptions to Figure 1A and Figure 6B. As detailed above, presentation of the results in boxplots or scatterplots is challenging due to large variation in the formation of deglycosylation shunt products. As such, the plots would either have been illogical if all

reactions were included or highly biased if only a selection of reactions were presented. We have added details of HPLC experimental profiles to the enzymatic activity assays in the Methods sections (the details were already present in the section describing analysis of culture extracts). Information on wavelength and HPLC column temperature is included in the revised manuscript Methods section.

Reviewer #2 (Remarks to the Author):

- What are the noteworthy results?

1. The natural redox partners ferredoxin FDX4 and ferredoxin reductase FDR3 were identified through transcriptomic analysis, which are crucial for DoxA catalysis.
2. Structural analysis and density functional theory (DFT) calculations revealed that the inefficient C-14 hydroxylation of daunorubicin is due to the unfavorable anti-conformation of its methyl ketone sidechain.
3. Advances in rational strain engineering led to a 180% increase in doxorubicin yields, improving the production profile.

- Will the work be of significance to the field and related fields? How does it compare to the established literature? If the work is not original, please provide relevant references.

There is some progress, but it is not significant. The main achievement is obtaining the three-dimensional structures of the complexes formed by the DoxA enzyme and the substrates DNR, DHD, and DOD.

- Does the work support the conclusions and claims, or is additional evidence needed?

Additional evidences are needed. For example, high-resolution three-dimensional structure of apo DoxA, and direct use of DNR as substrate to verify the product inhibition of DXR, etc.

- Are there any flaws in the data analysis, interpretation and conclusions? Do these prohibit publication or require revision?

Yes, I have attached a file. I believe these should prohibit publication.

- Is the methodology sound? Does the work meet the expected standards in your field?

No, I do not believe the methodology is sound, and the work does not meet the expected standards in the field.

- Is there enough detail provided in the methods for the work to be reproduced?

The substrate DOD is not readily available. Additionally, reproducing the other data as a whole is relatively straightforward.

RESPONSE: Thank you for the overall evaluation. We have addressed the concerns of the reviewer below in the appropriate sections describing the specific issues that were raised. We would especially grateful to the reviewer for noting inconsistencies and typographical errors in our use of the abbreviations FDX/FDR/PDX/PDR. In this revised version, we have reverted to a more suitable nomenclature, where first the species is mentioned (Sp for *Streptomyces peucetius* and So for *Spinacia oleracea*) followed by abbreviation of the enzyme (Fdx or FdR for ferredoxin or ferredoxin reductase, respectively). However, we have retained the original abbreviations Pdx and PdR for enzymes from *Pseudomonas putida*, since these originate from the words “putidaredoxin” and “putidaredoxin reductase”, respectively.

- Is the CC (1/2) value for DoxA apo reasonable? It is significantly lower than 0.5. The Rpim value is 0.902, which is much greater than 0.5. The resolution is too low, resulting in lower reliability for the overall apo structure.

RESPONSE: We agree that improved statistics for this apo dataset would be preferable. However, it is likely that this reflects a relatively flexible or unstructured protein. In the original submission, we included the

dataset to address minor issues, particularly to highlight flexibility in the F-G loop, which is well-known for P450 enzymes. The B-factors of the other structures already indicate this sufficiently.

We concur with the reviewer that the dataset quality does not meet the necessary standard. Because the structure does not provide additional mechanistic insight, we have removed it from the manuscript and instead emphasized the B-factors and the F–G loop to clarify the proposed substrate entry pathway.

- Is "Na₂SO₄" written incorrectly in Figure S3? Additionally, is the formatting consistent with academic standards?

RESPONSE: Agreed and amended. We have corrected this typo in the revised manuscript.

- The compound 7-deoxydaunoribicinone does not appear anywhere in the main text of the manuscript.

RESPONSE: We did not show the chemical structure of 7-deoxydaunoribicinone in the original submission as it is a well-known shunt product. In the revised manuscript, we discuss the formation of the compound in more detail and the name "7-deoxydaunoribicinone" (line 145). In addition, the chemical structure is shown in Figure S4 together with the NMR data of the molecule (Figures S5-S10).

- Line 67: V_{max} values writing specification?

RESPONSE: We sought to avoid including exact V_{max} values for previously conducted studies since the conditions in enzyme assays have a substantial influence on the values. This point is discussed in the introduction, and we elected to use fold difference to relay conversion rates of the various DoxA substrates. Furthermore, the previous studies were conducted in the absence of DnrV, which we show here to have a significant influence on the catalytic values.

- Line 143-145: The manuscript states that "FDX4 is a 2Fe-2S ferredoxin similarly to SFX and PDR," but later concludes that FDX4 is a Rieske-type ferredoxin while PDR is classified as a non-Rieske [2Fe-2S] ferredoxin (Figure S3).

RESPONSE: Thank you for point out these errors in the fluency of the text. In the revised manuscript, we state that Fdx4 belongs to a class of Rieske enzymes and further describe known members of the family based on evidence supported by a multiple sequence alignment (new Figure S11). We also explain how Fdx4 differs from SoFdx and Pdx (line 152). We also explain that each of these ferredoxins contain [2Fe-2S] clusters, but that the coordinating amino acids residues are unique in FDX4 (2 x His, 2 x Cys) compared to PDX/FDX (4 x Cys).

This raises two concerns:

1. Logical inconsistency: If FDX4 shares similarity with PDR, their classifications as Rieske-type and non-Rieske ferredoxins, respectively, appear contradictory.

RESPONSE: Agreed and amended. The low sequence similarity does not warrant comparison of the sequences of SpFdx4 and Pdx and we have revised the text accordingly.

2. Functional misclassification: The description of PDR as a ferredoxin may require re-evaluation. Ferredoxin reductases (e.g., PDR family proteins) and ferredoxins (e.g., SFX family) are functionally distinct redox partners, not homologous proteins. This mischaracterization undermines the comparative analysis.

RESPONSE: Agreed and amended. We noticed two typographical errors where the putidaredoxin was incorrectly abbreviated as PDR instead of PDX.

- Line 149-150:

The statement that "Five of the six FDR proteins related to PDR were produced in *E. coli*" (R1, R2, R3, R4, R7) raises methodological concerns:

1. Unclear selection criteria: The manuscript does not define which six FDR proteins are considered "related to PDR" or specify the criteria for their selection. The phylogenetic tree (Figure 3B) does not clearly demarcate a clade containing exactly six FDR proteins. For instance, the purple-clade region (putative PDR-related group) appears to contain more than six members. The rationale for excluding one protein from heterologous expression requires justification. Was this due to failed expression/purification, or evolutionary divergence?

RESPONSE: Thank you for noticing these inconsistencies. We have corrected the text in the revised manuscript, which now reads "Five of the seven FdR proteins". Each of the seven genes homologous to ferredoxin reductases were cloned, but only five of the proteins could be produced in soluble form in *E. coli*. The selection criteria including protein expression is described in the revised manuscript (line 160).

2. Phylogenetic inconsistencies:

The claimed PDR-related FDRs (R1, R4, R7) cluster within a subclade distinct from other FDRs, yet R2 and R3 appear evolutionarily distant from this group. This contradicts the implied functional relationship. If evolutionary proximity to PDR is the basis for selection, the six candidates should be explicitly annotated on the phylogenetic tree with bootstrap support values.

RESPONSE: Thank you for your valuable feedback. In response to your comments, we have performed a new phylogenetic analysis to provide a more accurate and robust representation of the relationships among the uncharacterized ferredoxin reductase proteins from *Streptomyces peucetius*.

To this end, we have constructed a new phylogenetic tree incorporating a broader and more representative set of reference sequences from established ferredoxin reductase families. This approach enabled us to more precisely determine the evolutionary relationships and assign the functions of the *Streptomyces* proteins with high confidence. This revised analysis has resulted in a well-resolved and statistically supported tree, as evidenced by high bootstrap values at the key nodes defining the major clades.

The revised tree (Figure 3B) clearly shows the three distinct evolutionary and functional families of ferredoxin reductases:

- Plant-type (pink): This clade represents the plant-type ferredoxin reductases, which are typically involved in photosynthesis. The absence of this protein type in our *Streptomyces* species is consistent with its non-photosynthetic, heterotrophic metabolism.
- Mitochondrial-type (blue): This clade contains the mitochondrial-type ferredoxin reductases (adrenodoxin reductase-like) and includes our *Streptomyces* proteins, SpFdr2 and SpFdr3. This grouping is consistent with their likely role in supporting P450-mediated metabolic pathways.
- Putidaredoxin Reductase-like (purple): This clade contains the putidaredoxin reductase-like ferredoxin reductases, which forms a distinct family. The remaining five *Streptomyces* proteins cluster robustly within this group, indicating their evolutionary and functional relationship to this enzyme family.

- Enzyme activity assay section: In Fig. 4A, using DOD as the substrate, only a small amount of DXR is produced at a DnrV concentration of 10 μ M, while at 20 μ M, the yield of DXR is somewhat higher, but still lower than that of DNR. However, in Fig. 4C, when examining the effect of DXR concentration on the reaction, DOD is again used as the substrate, and at a DnrV concentration of 10 μ M, the amount of DXR

produced in the absence of added DXR is significantly higher than in Fig. 4A, even exceeding the yield of DNR. This presents a contradiction, as indicated by the red box in the figure below.

RESPONSE: We agree that inconsistencies in product yields are present in different figures in the activity assays. As explained to Reviewer #1, these fluctuations are due to the challenges in establishing a robust enzymatic assays system; the development and optimization of the system took two years. We have observed variation in the activities of the enzymes both between different purification batches, depending on storage time in the freezer. In addition, even minor changes such as a new vial of the co-substrate NADP⁺ or new stocks for catalase for removal of H₂O₂ or glucose dehydrogenase for regeneration of NADPH had an influence in the product yields. However, all of the data presented in the various manuscript figures are derived from experiments that were conducted at the same time to ensure consistency and scientific rigor.

• Line 168-170:

Why was DOD used as the substrate in the experiment examining the effect of DXR concentration on the reaction, rather than DHD or DNR? How can we determine whether the inhibition of the DoA reaction is due to the DXR product or if DoxA exhibits catalytic region selectivity?

RESPONSE: Thank you for the comment. DOD was used as a substrate because it has the highest relative turnover rate for the three DoxA-catalyzed reactions. We reasoned that use of this substrate would provide the clearest signal for decreased product formation upon titration with DXR. DHD could have been another

feasible option since the conversion of DHD to DNR is high. DNR was not selected as a substrate due to the relatively poor reaction turnover, and the significant data variance between individual samples.

Our hypothesis for DoxA product inhibition is supported by the observation that DoxA binds DXR (see Figure 4E). We do not observe that product inhibition and catalytic C-H oxidation regio-selectivity are mutually exclusive in the case of DoxA. The reaction that is most interesting from an applied industrial perspective is the conversion of DNR to DXR, which does not occur in the wild type *S. peucetius* strain. Therefore, it is apparent that wt DoxA has high catalytic selectivity for DOD and DHD, but not for DNR.

- Line 400: 16 mg mL⁻¹. Pay attention to format specifications.

RESPONSE: Agreed and amended.

Reviewer #3 (Remarks to the Author):

This is an excellent study on engineering a P450 system for improved production of doxorubicin, an important chemotherapeutic agent. The authors have improved catalysis by finding the natural redox partners for DoxA and by using an anthracycline binding protein to block product inhibition of the P450. I have only a few minor comments.

RESPONSE: Thank you for the overall evaluation.

1. Somewhere in the paper the author should give the correct CYP name for DoxA.

RESPONSE: Agreed and amended. We now refer to DoxA as “a heme containing cytochrome P450” enzyme (line 65).

2. In the figure legend to figure 4 they refer to panel “f” as “e”.

RESPONSE: Agreed and amended.

3. The MD doesn't add much especially using compound I. The lifetime of compound I is on the order of picoseconds, as estimated from radical clocks, so a 1000 ns simulation really is not of much relevance. A better model would be the oxy-complex. There are a few good P450-oxy-complex structures as a basis for modeling.

RESPONSE: While the oxy-complex persists across a longer timeframe within the P450 reaction cycle, it is the heme iron-oxo (compound I) that is the primary reactive oxygen species responsible for the C-H hydrogen atom abstraction that leads to substrate oxidation (Munro et al, 2013; Shaik et al, 2010). In addition, while 1000ns is a much longer timescale than the presence of the iron-oxo experimentally, (Munro et al, 2013) there is ample precedent for using long-scale simulations with the iron-oxo species to model P450 reactivity (Caddell Haatveit et al, 2019; Shende et al 2023; Narayan et al 2015). Therefore, we believe that use of long-scale simulations of substrates in complex with compound I is appropriate for this study.

4. In the MD word they only report the percentage of frames where C14 is in position for hydroxylation. They should also report the percentage of frames for C13.

RESPONSE: Response: Thank you for this suggestion, we have repeated our MD simulations in triplicate and report these values in revised Table S5 as noted above.

REFERENCES:

Caddell Haatveit, K.; Garcia-Borràs, M.; Houk, K. N. Computational Protocol to Understand P450 Mechanisms and Design of Efficient and Selective Biocatalysts. *Front. Chem.* **2019**, *6*, 663. <https://doi.org/10.3389/fchem.2018.00663>.

Fisher, J., Abdella, B. R. J. & McLane, K. E. Anthracycline antibiotic reduction by spinach ferredoxin-NADP+ reductase and ferredoxin. *Biochemistry* *24*, 3562–3571 (1985). <https://doi.org/10.1021/bi00335a026>

Munro, A. W.; Girvan, H. M.; Mason, A. E.; Dunford, A. J.; McLean, K. J. What Makes a P450 Tick? *Trends Biochem. Sci.* *2013*, *38* (3), 140–150. <https://doi.org/10.1016/j.tibs.2012.11.006>.

Narayan, A. R. H.; Jiménez-Osés, G.; Liu, P.; Negretti, S.; Zhao, W.; Gilbert, M. M.; Ramabhadran, R. O.; Yang, Y.-F.; Furan, L. R.; Li, Z.; Podust, L. M.; Montgomery, J.; Houk, K. N.; Sherman, D. H. Enzymatic Hydroxylation of an Unactivated Methylene C–H Bond Guided by Molecular Dynamics Simulations. *Nat. Chem.* *2015*, *7* (8), 653–660. <https://doi.org/10.1038/nchem.2285>.

Shaik, S.; Cohen, S.; Wang, Y.; Chen, H.; Kumar, D.; Thiel, W. P450 Enzymes: Their Structure, Reactivity, and Selectivity—Modeled by QM/MM Calculations. *Chem. Rev.* *2010*, *110* (2), 949–1017. <https://doi.org/10.1021/cr900121s>.

Shende, V. V.; Harris, N. R.; Sanders, J. N.; Newmister, S. A.; Khatri, Y.; Movassaghi, M.; Houk, K. N.; Sherman, D. H. Molecular Dynamics Simulations Guide Chimeragenesis and Engineered Control of Chemoselectivity in Diketopiperazine Dimerases. *Angew. Chem. Int. Ed.* n/a (n/a). <https://doi.org/10.1002/anie.202210254>.